# Specialization of the chromatin remodeler RSC to mobilize partially-unwrapped nucleosomes

**Alisha Schlichter†, Margaret M Kasten†, Timothy J Parnell, Bradley R Cairns***

Howard Hughes Medical Institute (HHMI), Department of Oncological Sciences, Huntsman Cancer Institute, University of Utah School of Medicine, Salt Lake City, United States

**Abstract** SWI/SNF-family chromatin remodeling complexes, such as *S. cerevisiae* RSC, slide and eject nucleosomes to regulate transcription. Within nucleosomes, stiff DNA sequences confer spontaneous partial unwrapping, prompting whether and how SWI/SNF-family remodelers are specialized to remodel partially-unwrapped nucleosomes. *RSC1* and *RSC2* are orthologs of mammalian *PBRM1* (polybromo) which define two separate RSC sub-complexes. Remarkably, in vitro the Rsc1-containing complex remodels partially-unwrapped nucleosomes much better than does the Rsc2-containing complex. Moreover, a *rsc1Δ* mutation, but not *rsc2Δ*, is lethal with histone mutations that confer partial unwrapping. Rsc1/2 isoforms both cooperate with the DNA-binding proteins Rsc3/30 and the HMG protein, Hmo1, to remodel partially-unwrapped nucleosomes, but show differential reliance on these factors. Notably, genetic impairment of these factors strongly reduces the expression of genes with wide nucleosome-deficient regions (e.g., ribosomal protein genes), known to harbor partially-unwrapped nucleosomes. Taken together, Rsc1/2 isoforms are specialized through composition and interactions to manage and remodel partially-unwrapped nucleosomes.

**\*For correspondence:**
brad.cairns@hci.utah.edu

†These authors contributed equally to this work

**Competing interests:** The authors declare that no competing interests exist.

## Introduction

Nucleosomes regulate transcription in diverse ways and can either block or attract transcriptional regulators (*Workman and Kingston, 1998*; *Iyer, 2012*). At promoters, nucleosome positioning and/or occupancy plays a central role in regulating transcription factor binding, with transitions in nucleosome positioning typically accompanying activation. SWI/SNF-family ATP-dependent chromatin remodeling complexes (CRCs) have evolved to conduct nucleosome sliding and ejection, and enable transcription factor access to DNA. These CRCs are complex in both composition and mechanism; they utilize a catalytic ATPase to translocate DNA around nucleosomes to conduct nucleosome sliding and eviction, and contain an additional set of proteins to help target and regulate each complex (*Clapier and Cairns, 2009*; *Lorch and Kornberg, 2017*; *Narlikar et al., 2013*).

The SWI/SNF-family remodeler, RSC (Remodels the Structure of Chromatin), from the budding yeast *S. cerevisiae,* is both essential and abundant, and has long served as a prototype CRC. RSC complex (like others in the SWI/SNF family) is found in more than one compositional subtype, and contains either Rsc1 or its highly-related paralog, Rsc2 (*Cairns et al., 1999*). Rsc1 and Rsc2 are orthologs of the mammalian polybromo, as both contain multiple bromodomains, a bromodomain-adjacent homology (BAH) domain, and a DNA binding motif (AT Hook or HMG box). Additional RSC compositional variation has been suggested, involving the association of two additional paralogous RSC subunits, Rsc3 and Rsc30 (*Campsteijn et al., 2007*; *Chambers et al., 2012*), which are zinc cluster DNA-binding proteins with affinity for GC-rich sequences (*Badis et al., 2008*). *RSC1* and *RSC2* are redundant for viability (*rsc1Δ rsc2Δ* mutants are inviable), however loss of only one confers

mild but dissimilar phenotypes, suggesting overlapping essential functions alongside limited unique functions (*Baetz et al., 2004*; *Chambers et al., 2012*; *Cairns et al., 1999*; *Bungard et al., 2004*). To gain further understanding regarding the roles of RSC1 and RSC2 complexes, we characterized Rsc1- and Rsc2-containing complexes through purification and in vitro biochemical assays, alongside in vivo genetic and genomic characterization. Both approaches converged to reveal an interplay of functional roles for Rsc1/2, Rsc3/30 and a partner high-mobility group (HMG) domain protein, Hmo1, in managing partially-unwrapped nucleosomes.

Partially-unwrapped nucleosomes are defined here as those in which the DNA has released (or displays a tendency to release) from the histone octamer at one or both of the symmetric locations where DNA enters/exits the nucleosome, while the central DNA gyre maintains association with the octamer. DNA sequences differ in their affinity for histone octamers and propensity to form nucleosomes (*Anderson et al., 2002*); stiff homopolymer AT tracts deter nucleosome formation (*Segal and Widom, 2009*) whereas those with short (5 bp) alternating AT and GC tracts more easily adopt nucleosomal curvature, providing a lower cost in energy for nucleosome formation. The commonly used '601' nucleosome positioning sequence is synthetic, was selected for high affinity (*Lowary and Widom, 1998*), and displays a 5 bp AT/GC alternating pattern. In contrast, the *5S* rRNA gene sequence is naturally occurring and of lesser affinity (comparable to genome averages) in comparison to 601 (*Polach and Widom, 1995*; *Dong et al., 1990*; *Li and Widom, 2004*; *Zhou et al., 2019*; *Mauney et al., 2018*). Here, there is some debate whether the entry/exit DNA ends of the *5S* positioning sequence displays higher or lower rates of detachment from the octamer than the 601 sequence (*North et al., 2012*; *Chen et al., 2014*; *Zhou et al., 2019*). However, recent work using small-angle X-ray scattering (SAXS) with salt titration to compare the unwrapping dynamics of the *5S* and 601 nucleosomes demonstrates that the *5S* nucleosome unwraps more rapidly and at lower salt concentrations than the 601 nucleosome (*Mauney et al., 2018*; *Chen et al., 2014*).

Prior work has revealed the presence of partially-unwrapped 'fragile' nucleosomes at promoters with an especially wide nucleosome deficient region (NDR), and co-incidence of RSC, including at the majority of ribosomal protein genes (RPGs) (*Kubik et al., 2015*; *Brahma and Henikoff, 2019*; *Knight et al., 2014*). Indeed, in silico size fractionation of RSC-bound nucleosomes provides evidence that RSC occupies wrapped nucleosomes at the +1 and −1 promoter positions, as well as a partially-unwrapped nucleosome interposed between those two displaying lower/fractional occupancy – the basis for the appearance of a large NDR (*Brahma and Henikoff, 2019*). As partially-unwrapped nucleosomes are likely conformationally diverse (*Bilokapic et al., 2018*), RSC may have evolved to both recognize and remodel these nucleosomes. Prior work with the Rsc2-containing form of RSC revealed altered remodeling outcomes for partially-unwrapped (H3 R40A) nucleosomes (*Somers and Owen-Hughes, 2009*), further questioning whether an alternative form of RSC might better manage them. RSC mobilization of partially-unwrapped nucleosomes may allow sets of transcription factors regulated access to a section of DNA without keeping the region constitutively nucleosome-free – which may provide regulatory benefits and help maintain genome stability (see Discussion). Here, we provide multiple lines of evidence that a set of proteins in RSC (Rsc1, Rsc3/30), and interacting with RSC (Hmo1), cooperate to help remodel partially-unwrapped nucleosomes in vitro and in vivo.

## Results

### The Rsc3/30 heterodimer preferentially associates with the RSC1 complex via the CT2 domain

Rsc1 and Rsc2 are highly similar proteins (45% identical, 62% similar), with high homology present in the bromodomains, BAH domain, and the CT1 region – whereas the CT2 region, which is required for Rsc1 or Rsc2 assembly into RSC, is considerably more divergent (*Cairns et al., 1999*). We began by exploring whether these Rsc1- or Rsc2-containing subtypes differ in composition, beyond Rsc1/2 themselves. To determine, we purified RSC sub-complexes using a TAP tag (*Puig et al., 2001*) on either Rsc1 or Rsc2, which revealed Rsc3/30 at apparent stoichiometric levels in the Rsc1-containing complex, but substoichiometric levels in the Rsc2-containing complex (*Figure 1A and B*; *Chambers et al., 2012*). Furthermore, when purifications were conducted under increasing salt

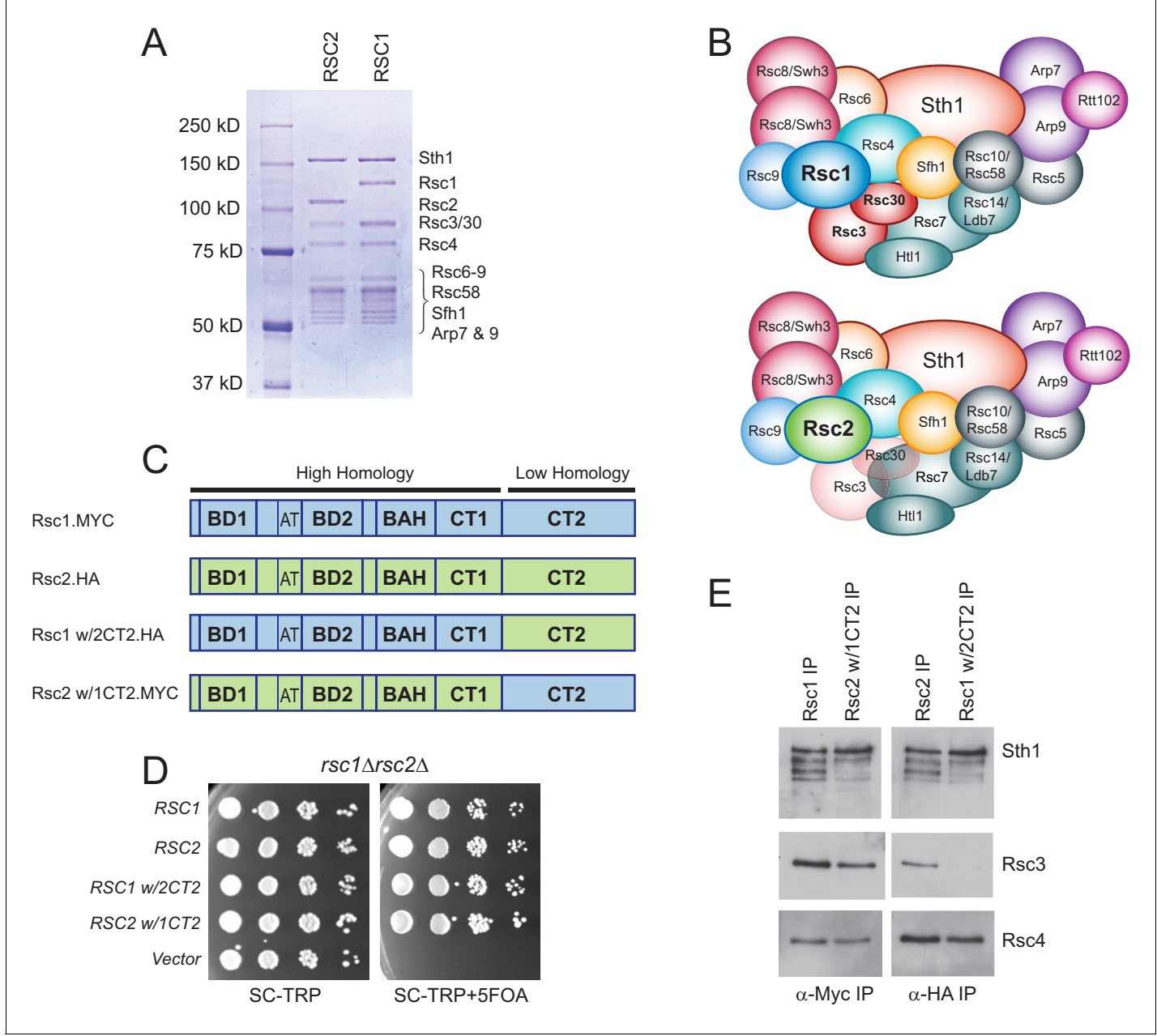

**Figure 1.** Rsc3/30 module has higher avidity for the RSC1 complex. (**A**) Purified RSC1-TAP and RSC2-TAP complexes (2 μg) analyzed on 7.5% SDS-PAGE gel stained with Coomassie dye. (**B**) RSC1 and RSC2 complex compositions, with decreased opacity displaying the reduced association of the Rsc3/30 module in the RSC2 complex. (**C**) Domain structure and swaps of Rsc1 and Rsc2. (**D**) CT2 domain swaps complement for viability. *TRP1*-marked plasmids bearing *RSC1* (p609), *RSC2* (p604), *RSC1 w/2CT2* (p3097), *RSC2 w/1CT2* (p3098), or vector (pRS314) were transformed into *rsc1Δ rsc2Δ* [*RSC1.URA3*] (YBC800), and spotted as 10x serial dilutions to SC-TRP or to SC-TRP+5FOA to force the loss of the *RSC1.URA3* plasmid. ' w/' indicates 'with'. One of four biological replicates shown. (**E**) The Rsc3/30 module associates more strongly with the CT2 region of Rsc1. Immunoprecipitations of Rsc1, Rsc2 w/1CT2, Rsc2, Rsc1 w/2CT2 from whole cell extracts. Blots were probed with anti-Sth1, then stripped and reprobed for anti-Rsc3 and anti-Rsc4. One of three technical replicates shown. *Figure 1—figure supplement 1*. The Rsc3/30 module associates with the RSC1 complex at high stringency. *Figure 1—figure supplement 2*. Additional swaps and truncations define the region of Rsc3/30 association.

The online version of this article includes the following figure supplement(s) for figure 1:

**Figure supplement 1.** The Rsc3/30 module associates with the RSC1 complex at high stringency.

**Figure supplement 2.** Additional swaps and truncations define the region of Rsc3/30 association.

conditions, Rsc3/30 association with Rsc1-TAP was maintained, but was lost with Rsc2-TAP (*Figure 1—figure supplement 1*) confirming higher avidity of Rsc3/30 for the Rsc1 sub-type.

To identify whether a particular region within Rsc1 mediates this preferential association with Rsc3/30, we performed domain swaps between Rsc1 and Rsc2 (*Figure 1C*) and checked for complementation and Rsc3 association. To assess this, a *rsc1Δ rsc2Δ* strain containing *RSC1* on a *URA3*-marked plasmid was transformed with *TRP1*-marked plasmids containing Rsc1/2 domain swap derivatives, and complementation was demonstrated by their ability to lose the *URA3*-marked *RSC1* plasmid with 5FOA (*Figure 1D*). Here, co-immunoprecipitation revealed strong Rsc3 association with Rsc1, and with Rsc2 derivatives only if they contained the CT2 region of Rsc1 (*Figure 1E*). Conversely, strong Rsc3 association with Rsc1 was lost when Rsc1 contained the CT2 region of Rsc2. Similar approaches involving internal Rsc1/2 deletions and additional swap positions provided further refinement, narrowing the region on Rsc1 responsible for strong Rsc3/30 association to residues 617–777 (*Figure 1—figure supplement 2A–D*). Several structures of RSC-nucleosome complexes have recently been published (*Wagner et al., 2020*; *Ye et al., 2019*; *Patel et al., 2019*). However, as Rsc1, Rsc2, Rsc3, and Rsc30 are within the flexible regions, these structures contain only partial models (or lack density/models). While the Rsc1-Rsc3/30 interaction has not been resolved structurally, crosslinks were observed between Rsc2 CT2 and Rsc3/30 heterodimer (*Wagner et al., 2020*), an interaction we find to be much stronger in Rsc1.

## The RSC1 complex slides 5S nucleosomes better than the RSC2 complex

Having defined and isolated the four main RSC subtypes (RSC1 or RSC2, +/- Rsc3/30; *Figure 2A*), we then tested for differences in ATPase activity, and remodeling efficiency, by examining nucleosomes of typical wrapping/affinity, such as those formed with the sea urchin *5S* DNA nucleosome positioning sequence (NPS) and recombinant yeast histones. First, all four RSC complexes displayed similar DNA-dependent ATPase activities in typical $V_{max}$ determinations with plasmid DNA (*Figure 2B*). The activity of one complex, RSC1-3/30, plateaued earlier than the other RSC complexes, but otherwise the two complexes behaved similarly. However, major differences were observed with *5S* nucleosomes – RSC1 complex displayed much greater sliding activity than RSC2 (*Figure 2C* and *Figure 2—figure supplement 1* for quantification). As shown previously for RSC, all sliding activities with RSC1 and RSC2 complexes are ATP dependent (*Cairns et al., 1996* and *Figure 2—figure supplement 2A*). RSC1 and RSC2 complexes both bound *5S* nucleosomes comparably in the absence of ATP (*Figure 2—figure supplement 2B*) indicating that their differences in activities occur after the initial engagement of the *5S* nucleosome. Finally, the Rsc3/30 module did not affect their ATPase activities (*Figure 2B*) nor their binding to *5S* nucleosomes (*Figure 2—figure supplement 2B*); however, in both the RSC1 and RSC2 complexes, Rsc3/30 inclusion moderately inhibited remodeling (*Figure 2C*). Taken together, RSC1 complex displays markedly higher sliding activity on *5S* nucleosomes compared to RSC2 complex.

We note that the RSC-remodeled *5S* nucleosomal product migrates more slowly on native gels than the starting nucleosome in our sliding assay. This could result from an altered position of the octamer along the DNA, from the partial unwrapping of DNA from the octamer, or possibly from the creation of a hexasome during the remodeling reaction (which would create a more 'open' and unwrapped structure). To determine whether the slower-migrating RSC remodeled product was the result of H2A/H2B dimer loss from the nucleosome, we assembled 174 bp *5S* nucleosomes using yeast octamers that were fluorescently labeled with Oregon Green (OG) on H2A (Q114C) and conducted sliding assays with RSC1-3/30 (*Figure 2—figure supplement 3A*). The ratio of H2A to DNA was calculated for both the 'start' and 'slid' bands at each time point, normalized to the starting nucleosomal band (*Figure 2—figure supplement 3B*). Here, the 'slid' band maintained a dimer to DNA ratio similar to the starting nucleosome, rather than that predicted for a hexasome or tetrasome, supporting the identity of the slower-migrating band being an intact nucleosome. We will, therefore, refer to this band on the native gel as the 'slid' nucleosome position.

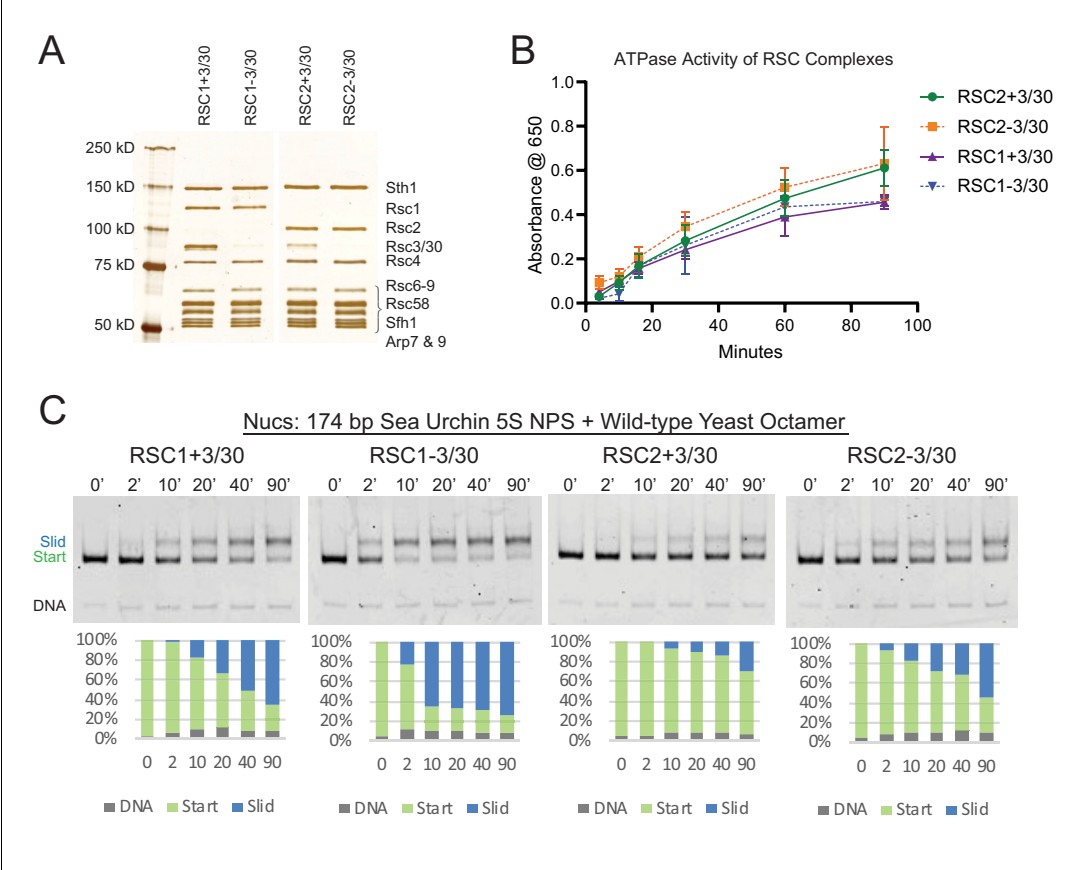

**Figure 2.** RSC1 and RSC2 complexes differ in remodeling activity on a sea urchin *5S* mononucleosomal substrate. (**A**) Alternative RSC1 and RSC2 complexes, with the Rsc3/30 module maintained or removed during purification. Purified RSC complexes (600 ng) were analyzed on a 6% polyacrylamide SDS-PAGE gel stained with silver. The RSC1 and RSC2 complexes are from the same gel, but were moved adjacent for the depiction. (**B**) ATPase time course of RSC1 and RSC2 with and without the Rsc3/30 module. Values are the mean +/- standard deviation from two separate RSC preps for each RSC complex assayed in triplicate. (**C**) Comparative sliding of 174 bp sea urchin *5S* yeast mononucleosomes (20 nM) by RSC1 and RSC2 complexes (30 nM). The nucleosomal Start (green), Slid (blue), and free DNA (grey) bands were quantified and reported as a percent of the total signal. The online version of this article includes the following source data and figure supplement(s) for figure 2:

**Source data 1.** ATPase time course data.
**Source data 2.** RSC sliding of wt *5S* yNucs.
**Figure supplement 1.** Quantification of RSC sliding 174 bp *5S* wild-type yeast mononucleosomes as conducted in *Figure 2C*.
**Figure supplement 1—source data 1.** Quantification of RSC sliding *5S* wt yNucs.
**Figure supplement 2.** RSC complexes without Rsc3/30 module are ATP-dependent chromatin remodelers and RSC1 and RSC2 complexes bind similarly to 174 bp sea urchin *5S* mononucleosomes.
**Figure supplement 3.** The RSC *5S* sliding product is not the result of H2A/H2B dimer loss.
**Figure supplement 3—source data 1.** Quantification of RSC1 sliding H2A-OG *5S* yNucs.

## *rsc1∆* mutation is lethal in combination with histone mutations that confer partial unwrapping

In principle, a variety of factors might underlie better relative remodeling by RSC1 complexes on *5S* nucleosomes. To provide insight into Rsc1/2 differences and to help guide further in vitro approaches, we conducted unbiased genetic screens to identify histone mutations that are selectively lethal with *rsc1∆* or *rsc2∆* mutations. Here, we utilized an alanine scanning approach in which each of the histone residues is separately mutated to alanine. To implement, we created *rsc1∆* or *rsc2∆* deletions (separately) within a 'histone shuffle' strain (*rsc1∆* or *rsc2∆, h3-h4∆* [*H3-H4.URA3*]) and combined those with a library encoding all viable histone H3-H4 alanine substitutions on *TRP1*-marked plasmids (*Nakanishi et al., 2008*). We then assessed viability following forced loss of the wild-type *H3-H4* plasmid on 5FOA-containing medium. Histone mutations that are lethal with both

*rsc1Δ* and *rsc2Δ*, or lethal uniquely with *rsc2Δ*, were found distributed throughout the nucleosome. In striking contrast, those mutations that were lethal specifically with *rsc1Δ* mapped exclusively within the H3 αN helix region – the position where DNA enters/exits the nucleosome (*Table 1*, *Figure 3A–B*). Furthermore, the specific H3 αN helix mutations obtained in our screen overlap strongly with those reported to increase partial unwrapping using FRET formats (*Ferreira et al., 2007*), whereas alanine substitutions that did not show a phenotype (P43A, E50A, K56A) had little effect on FRET/unwrapping (*Table 2*). Given the results from the H3-H4 screen, we then performed a screen combining *rsc1Δ* or *rsc2Δ* with H2A-H2B mutations (*Nakanishi et al., 2008*). In keeping with the results above, we find *rsc1Δ*-specific synthetic lethality primarily with mutations in the H2A C-terminus which interact with the H3 αN helix, as well as with specific histone-DNA contacts (H2A R78 and H2A R30; *Figure 3—figure supplement 1*). Taken together, our genetic results, which encompass the entire

**Table 1.** Summary of Histone H3-H4 screen with *rsc1Δ* and *rsc2Δ*.

Library of *TRP1*-marked plasmids containing H3-H4 residues mutated to alanine were transformed into *h3-h4Δ* [*H3-H4.URA3*] (YBC1939), *rsc1Δ h3-h4Δ* [*H3-H4.URA3*] (YBC2090) or *rsc2Δ h3-h4Δ* [*H3-H4.URA3*] (YBC3040), and spotted to SC-TRP, or SC-TRP + 5FOA to force the loss of the WT histone plasmid and test for synthetic lethality. Histone mutations that were lethal on their own in WT RSC are shaded grey, lethal in combination with *rsc1Δ* are in bold and highlighted yellow, lethal with *rsc2Δ* are italicized and highlighted in blue, and residues that were lethal with both *rsc1Δ* and *rsc2Δ* are highlighted in green.

| H3 | 1 | 2 | 3 | 4 | 5 | 6 | 7 | 8 | 9 | 10 | 11 | 12 |
|---|---|---|---|---|---|---|---|---|---|---|---|---|
| A | R2A | T11A | L20A | T32A | R40A | R49A | S57A | L65A | E73A | L82A | L92A | V101A |
| B | T3A | G12A | S22A | G33A | Y41A | E50A | T58A | P66A | I74A | R83A | Q93A | S102A |
| C | K4A | G13A | K23A | G34A | K42A | I51A | E59A | F67A | Q76A | F84A | E94A | L103A |
| D | Q5A | K14A | R26A | V35A | P43A | R52A | L60A | Q68A | D77A | Q85A | S95A | F104A |
| E | T6A | P16A | K27A | K36A | G44A | R53A | L61A | R69A | F78A | S86A | V96A | E105A |
| F | R8A | R17A | S28A | K37A | T45A | F54A | I62A | L70A | K79A | S87A | E97A | D106A |
| G | K9A | K18A | P30A | P38A | V46A | Q55A | R63A | V71A | T80A | I89A | Y99A | T107A |
| H | S10A | Q19A | S31A | H39A | L48A | K56A | K64A | R72A | D81A | G90A | L100A | N108A |
| H3 | 1 | 2 | 3 | 4 | 5 | 6 | 7 | 8 | 9 | 10 | 11 | 12 |
| A | L109A | Q120A | R129A | | | | | | | | | |
| B | I112A | K121A | L130A | | | | | | | | | |
| C | H113A | K122A | R131A | | | | | | | | | |
| D | K115A | D123A | G132A | | | | | | | | | |
| E | R116A | I124A | E133A | | | | | | | | | |
| F | V117A | K125A | R134A | | | | | | | | | |
| G | T118A | L126A | S135A | | | | | | | | | |
| H | I119A | R128A | | | | | | | | | | |
| H4 | 1 | 2 | 3 | 4 | 5 | 6 | 7 | 8 | 9 | 10 | 11 | 12 |
| A | S1A | G9A | H18A | I26A | R35A | K44A | E52A | F61A | S69A | R78A | V86A | R95A |
| B | G2A | L10A | R19A | Q27A | R36A | R45A | E53A | L62A | V70A | K79A | V87A | T96A |
| C | R3A | G11A | K20A | G28A | L37A | I46A | V54A | E63A | T71A | T80A | Y88A | L97A |
| D | G4A | K12A | I21A | I29A | R39A | S47A | R55A | S64A | Y72A | V81A | L90A | Y98A |
| E | K5A | G13A | L22A | T30A | R40A | G48A | V57A | V65A | T73A | T82A | K91A | G99A |
| F | G6A | G14A | R23A | K31A | G41A | L49A | L58A | I66A | E74A | S83A | R92A | F100A |
| G | G7A | K16A | D24A | P32A | G42A | I50A | K59A | R67A | H75A | L84A | Q93A | G101A |
| H | K8A | R17A | N25A | I34A | V43A | Y51A | S60A | D68A | K77A | D85A | G94A | G102A |

| Lethal w/*rsc1Δ* | Lethal w/*rsc2Δ* | Lethal w/*rsc1Δ* and *rsc2Δ* | Lethal w/WT |
|---|---|---|---|

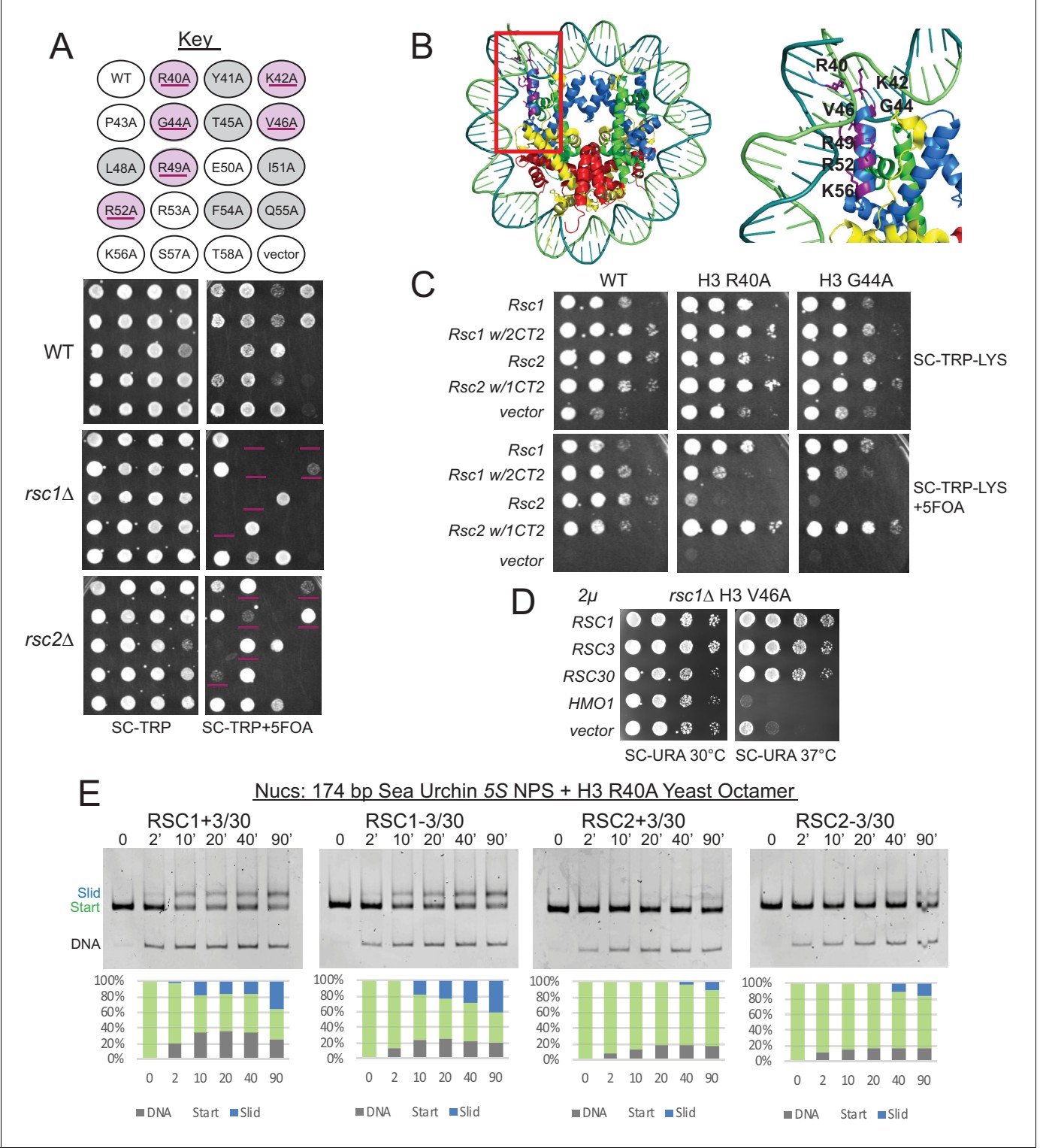

**Figure 3.** Mutations in the H3 αN helix are lethal in combination with *rsc1Δ*, but not *rsc2Δ*, and they reduce RSC remodeling of the *5S* nucleosome. (A) Histone H3 αN helix mutations that are lethal with *rsc1Δ*. *TRP1*-marked plasmids containing WT H4, and H3 mutations within the αN helix were transformed into *h3-h4Δ* [*H3-H4.URA3*] (YBC1939), *rsc1Δ h3-h4Δ* [*H3-H4.URA3*] (YBC2090) or *rsc2Δ h3-h4Δ* [*H3-H4.URA3*] (YBC3040), and spotted to SC-TRP or SC-TRP+5FOA to force the loss of the WT histone plasmid. Mutations that were lethal on their own without mutated RSC are shaded in grey. Mutations that were lethal in *rsc1Δ* but not *rsc2Δ* are shaded in purple and underlined. Transformants were grown at 30°C for 2 days. Shown is one of two biological replicates. (B) Location of the synthetic lethal *rsc1Δ* H3 αN helix mutations are depicted in purple on the nucleosome, PDB code 1ID3. *Figure 3 continued on next page*

*Figure 3 continued*

(C) The *RSC1 CT2* region complements the synthetic lethal *rsc1Δ* H3 αN helix mutations. *rsc1Δ rsc2Δ h3-h4Δ* [*RSC1.URA3*] with [*H3.WT, R40A,* or *G44A-H4.WT. LYS2*] (YBC3466, YBC3444, YBC3433) transformed with *TRP1*-marked plasmids bearing *RSC1* (p609), *RSC2* (p604), *RSC1 w/2CT2* (p3097), *RSC2 w/1CT2* (p3098), or vector (pRS314) and spotted as 10x serial dilution to SC-TRP-LYS, or SC-TRP-LYS+5FOA. Shown is one of four biological replicates. (D) High-copy *RSC3* or *RSC30* will partially suppress the Ts phenotype of *rsc1Δ H3V46A*. Strain *rsc1Δ h3-h4Δ* [*H3.V46A-H4.WT.TRP*] (YBC3586) transformed with *URA3*-marked high copy (2μ) plasmids containing *RSC1* (p705), *RSC3* (p1310), *RSC30* (p916), *HMO1* (p3390), or vector (pRS426), and spotted as 10x serial dilutions at 30°C or 37°C. Shown is one of two biological replicates. (E) Comparative sliding and ejection of 174 bp sea urchin *5S* NPS H3 R40A yeast mononucleosomes (20 nM) by RSC1 and RSC2 complexes (30 nM). The nucleosomal Start (green), Slid (blue), and free DNA (grey) bands were quantified and reported as a percent of the total signal.

The online version of this article includes the following source data and figure supplement(s) for figure 3:

**Source data 1.** RSC sliding *5S* R40A yNuc.
**Figure supplement 1.** Synthetic lethal mutations in each of the four histones H2A, H2B, H3, and H4 when combined with *rsc1Δ* or *rsc2Δ* mutations.
**Figure supplement 2.** Complementation by CT2 domain swaps when combined with H3 αN helix mutations.
**Figure supplement 3.** Quantification of RSC sliding 174 bp *5S* H3 R40A yeast mononucleosomes as conducted in *Figure 3E*.
**Figure supplement 3—source data 1.** Quantification of RSC sliding *5S* R40A yNucs.
**Figure supplement 4.** RSC1 and RSC2 complexes bind similarly to 174 bp sea urchin *5S* yeast mononucleosomes with the H3 R40A mutation.
**Figure supplement 5.** Mapping the positions of the wt and H3 R40A 174 bp sea urchin *5S* yeast mononucleosomes.
**Figure supplement 5—source data 1.** 174 bp *5S* nucleosome mapping.

nucleosome, strongly suggest that synthetic lethality in *rsc1Δ* strains is due to decreased ability of RSC2 complexes to remodel partially-unwrapped nucleosomes relative to RSC1 complexes.

## Increased Rsc3/30 association with RSC suppresses phenotypes associated with histone αN helix mutant combinations

We then explored whether the genetic differences observed with histone αN helix mutations in combination with *rsc1Δ* or *rsc2Δ* involve differential interaction of Rsc1/2 with Rsc3/30. First, we tested whether the CT2 domain of Rsc1, which interacts better with Rsc3/30 than its counterpart in Rsc2,

**Table 2.** *DNA end to end FRET measurements on mononucleosomes containing H3 αN helix mutations from *Ferreira et al., 2007* with the phenotype when combined with *RSC, rsc1Δ,* or *rsc2Δ*.

| Mutation | * FRET % | Phenotype |
|---|---|---|
| WT | 100 ± 6 | No phenotype |
| R40A | 71 ± 7 | Lethal w/*rsc1Δ* |
| Y41A | 72 ± 7 | Lethal w/WT |
| K42A | 54 ± 3 | Lethal w/*rsc1Δ* |
| P43A | 91 ± 3 | No phenotype |
| G44A | 68 ± 9 | Lethal w/*rsc1Δ* |
| T45A | 52 ± 5 | Lethal w/WT |
| V46A | 89 ± 5 | Lethal w/*rsc1Δ* |
| L48A | 86 ± 4 | Lethal w/WT |
| R49A | 67 ± 3 | Lethal w/*rsc1Δ* |
| E50A | 98 ± 6 | No phenotype |
| I51A | 81 ± 13 | Lethal w/WT |
| R52A | 78 ± 3 | Lethal w/*rsc1Δ* |
| R53A | 80 ± 4 | No phenotype |
| F54A | 96 ± 3 | Lethal w/WT |
| Q55A | 69 ± 11 | Lethal w/WT |
| K56A | 95 ± 4 | No phenotype |
| S57A | 102 ± 5 | No phenotype |
| K56Q | 82 ± 2 | Lethal w/*rsc1Δ* |

could confer growth when placed within a Rsc2 derivative. Here, growth was clearly restored to *rsc1Δ* H3 G44A and *rsc1Δ* H3 R40A combinations with a plasmid encoding Rsc1 or Rsc2 bearing the Rsc1 CT2, but not with Rsc2 or Rsc1 bearing the Rsc2 CT2 (*Figure 3C*). Additional domain swap experiments localize this complementation to the Rsc3/30 association region in Rsc1 (aa 617–777) (*Figure 3—figure supplement 2A*). As a complementary approach, we tested for *RSC3* or *RSC30* high-copy plasmid suppression. While high-copy *RSC3* or *RSC30* could not rescue *rsc1Δ* αN helix histone combined lethality (data not shown), suppression of the strong temperature sensitivity (Ts⁻) phenotype of *rsc1Δ* H3 V46A was observed (*Figure 3D*). Taken together, improving the association/functionality of Rsc3/30 suppresses phenotypes associated with αN helix histone mutations – further linking Rsc1/2 and Rsc3/30 to partial nucleosome unwrapping.

## Rsc1/2 differences are largely independent of bromodomain-histone interactions

We then tested the alternative hypothesis that the bromodomains of Rsc1/2 might interact differently with the main acetylated histone residue withinthe αN helix, H3 K56. However, the H3 K56A mutation was not lethal in combination with either *rsc1Δ* or *rsc2Δ* (*Figure 3A*). We then further tested H3 K56Q, which mimics the acetylated form. The K56Q mutation confers synthetic lethality when combined with *rsc1Δ*, but not with *rsc2Δ* (*Figure 3—figure supplement 2B*). Furthermore, domain swaps involving the highly homologous bromodomains and BAH domains between Rsc1 and Rsc2 did not confer phenotypic differences, nor did these swaps alter the synthetic lethality of *rsc1Δ* with histone αN helix mutations (*Figure 3—figure supplement 2A*). Notably, H3 K56A has little effect on DNA unwrapping, whereas K56Q promotes unwrapping (*Ferreira et al., 2007*; *Masumoto et al., 2005*) – and is lethal with *rsc1Δ*, further supporting an unwrapping function as being responsible for the phenotype. Thus, the lethality with *rsc1Δ* does not appear linked to bromodomains or histone acetylation, in agreement with findings that H3K56 acetylation does not enhance RSC binding (*Neumann et al., 2009*).

## RSC2 complexes are deficient in remodeling partially-unwrapped nucleosomes

Our genetic results prompted the examination of sliding by the RSC1 and RSC2 complexes on *5S* nucleosomes bearing a mutation (e.g., H3 R40A) predicted to confer partial unwrapping. Although sliding of nucleosomes bearing H3 R40A by either form of RSC is reduced relative to wild-type (WT) *5S* nucleosomes, RSC1 complexes were markedly more active than RSC2 complexes (*Figure 3E* and *Figure 3—figure supplement 3* for quantification), reinforcing the difference between RSC1 and RSC2. Additionally, both RSC1 and RSC2 complexes bind H3 αN helix mutant *5S* nucleosomes similarly (*Figure 3—figure supplement 4*), demonstrating that initial nucleosome binding is not inhibited by this octamer mutation, suggesting downstream remodeling activity as the affected step. To confirm and better define the extent of partial unwrapping observed on our 174 bp *5S* nucleosome with yeast octamers, we conducted ExoIII-S1 nuclease mapping (which removes DNA that is either outside of, or not well wrapped in a nucleosome; *Flaus, 2011*), and combined this with a high-throughput paired-end sequencing approach to define the endpoints and proportion of the nuclease-protected species. We found WT 174 bp *5S* yeast nucleosomes display a fully-wrapped side (position 158), and a side of partial unwrapping, in agreement with asymmetric *5S* nucleosome unwrapping previously shown (*Winogradoff and Aksimentiev, 2019*; *Chen et al., 2014*). Here, WT nucleosomes displayed a much higher proportion of largely-wrapped species (≥135 bp) than did H3 R40A nucleosomes (*Figure 3—figure supplement 5*). Since RSC2 is more deficient than RSC1 in repositioning H3 R40A nucleosomes, the mapping supports the hypothesis that RSC1 complexes manage partially-unwrapped nucleosomes better than RSC2 complexes. We note that H3 R40A *5S* nucleosomes are likely to have a distinct conformation that is not distinguished in our nuclease protection assay, since remodeling by both RSC1 and (more so) by RSC2 complexes is inhibited by the H3 αN helix mutation (compare *Figure 3E* with *Figure 2C*). These mutant octamers may enforce a greater degree of openness or distance between the *5S* DNA ends, as demonstrated previously (*Ferreira et al., 2007*), and thereby inhibit RSC activity (perhaps the transition of binding to DNA translocation, see Discussion) – a nucleosome perturbation better managed by RSC1 complexes.

## RSC cooperates with Hmo1 to remodel partially-unwrapped nucleosomes

Hmo1 is an HMGB family protein that stabilizes fragile/partially-unwrapped nucleosomes, particularly at rRNA and ribosomal protein gene promoters (*Hall et al., 2006*; *Panday and Grove, 2017*). Unlike other HMGB proteins which have an acidic CTD that promotes bending and nucleosome destabilization, Hmo1 is unique in containing a basic lysine rich C-terminal extension, and has been shown to stabilize chromatin and perform the functions of a linker histone (*Panday and Grove, 2016*). Hmo1 is proposed to bind near the nucleosome dyad and use its basic extension to bind linker DNA and prevent bending (*Panday and Grove, 2017*), which may also improve wrapping and thus cooperate with RSC1 or RSC2 to promote remodeling. We first tested for *rsc/hmo1* genetic interactions by combining *rsc1Δ* or *rsc2Δ* with *hmo1Δ*. Notably, we observe synthetic phenotypes in *rsc1Δ hmo1Δ* mutants, but not *rsc2Δ hmo1Δ* mutants (*Figure 4A*), suggesting a greater reliance of Rsc2 on Hmo1 for functional cooperativity. Furthermore, as we saw with the H3 αN helix mutations, this synthetic sickness was partially complemented by the presence of the Rsc1 CT2, and partially suppressed by high-copy *RSC3* or *RSC30*, providing further support that these proteins work together in a modular manner (*Figure 4B*).

To test RSC-Hmo1 associations in vivo, we performed co-immunoprecipitations between Rsc1 and Rsc2 with Hmo1. Hmo1 was endogenously tagged at its C-terminus with a V5 epitope in Myc-tagged Rsc1 and Rsc2 strains. Here, crosslinked chromatin extracts were prepared from log phase cells and sonicated, or treated with micrococcal nuclease, resulting primarily in mononucleosomes. Immunoprecipitation with anti-Myc or anti-V5 antibody followed by immunoblot analysis revealed that both Rsc1 and Rsc2 co-immunoprecipitate with Hmo1, which represents either a direct interaction between Hmo1 and RSC or colocalization on the same nucleosome(s) (*Figure 4C*, *Figure 4—figure supplement 1*). Taken together, Hmo1 shows physical interaction on chromatin with Rsc1 and Rsc2, but strong functional/genetic interaction primarily with Rsc2 (RSC2 complexes are reliant on Hmo1, in a *rsc1Δ* strain).

## Hmo1 strongly stimulates the sliding activity of Rsc2, and moderately stimulates Rsc1

Hmo1 stimulates the sliding activity of RSC and SWI/SNF complex in vitro on 601 nucleosomes (*Hepp et al., 2014*), but it is not known whether stimulation applies equally to RSC1 and RSC2 complexes, or has a differential effect on their action upon partially-unwrapped nucleosomes. One explanation for why *rsc1Δ hmo1Δ* mutants grow more poorly than *rsc2Δ hmo1Δ* mutants is that RSC1 is able to remodel those locations in the absence of Hmo1, but RSC2 is not. As Hmo1 enhances RSC remodeling activity (*Hepp et al., 2014*), we preincubated the *5S* nucleosomal template with increasing amounts of purified Hmo1 and tested for stimulation of RSC1/2 sliding activity. Interestingly, Hmo1 moderately stimulates RSC1 activity, whereas Hmo1 greatly stimulates RSC2 activity (*Figure 4D* and *Figure 4—figure supplement 2* for quantification), a property that is even more evident in conditions of limiting remodeler or at early time points (*Figure 4—figure supplement 3*). Moreover, both RSC1 and RSC2 slide the *5S* nucleosome at lower remodeler concentrations in the presence of Hmo1 (30 nM RSC without Hmo1 compare with 10 nM RSC with Hmo1), further supporting a role for Hmo1 in stimulating RSC remodeling activity.

## Well-wrapped nucleosomes are remodeled comparably by both RSC1 and RSC2 complexes

Beyond Hmo1 addition, we explored the ability of RSC1 and RSC2 complexes to remodel nucleosomes bearing a very strong positioning sequence: the optimized 601 sequence. Here, RSC1 and RSC2 complexes both displayed robust sliding activity with 601 nucleosomes, with RSC1 activity slightly higher (*Figure 4E* and *Figure 4—figure supplement 4* for quantification). We note a correlation here and in prior work (*Clapier et al., 2016*) between ejection and the use of strong positioning sequences – their initial stability from favored 'phasing' may convert to high instability and disfavored 'phasing' after 5 bp of DNA translocation – which may underlie the progressive ejection observed with 601 nucleosomes.

We then explored whether an extremely strong positioning sequence (601) might remain wrapped in the presence of an αN helix mutation and rescue RSC2 remodeling. Indeed, we found

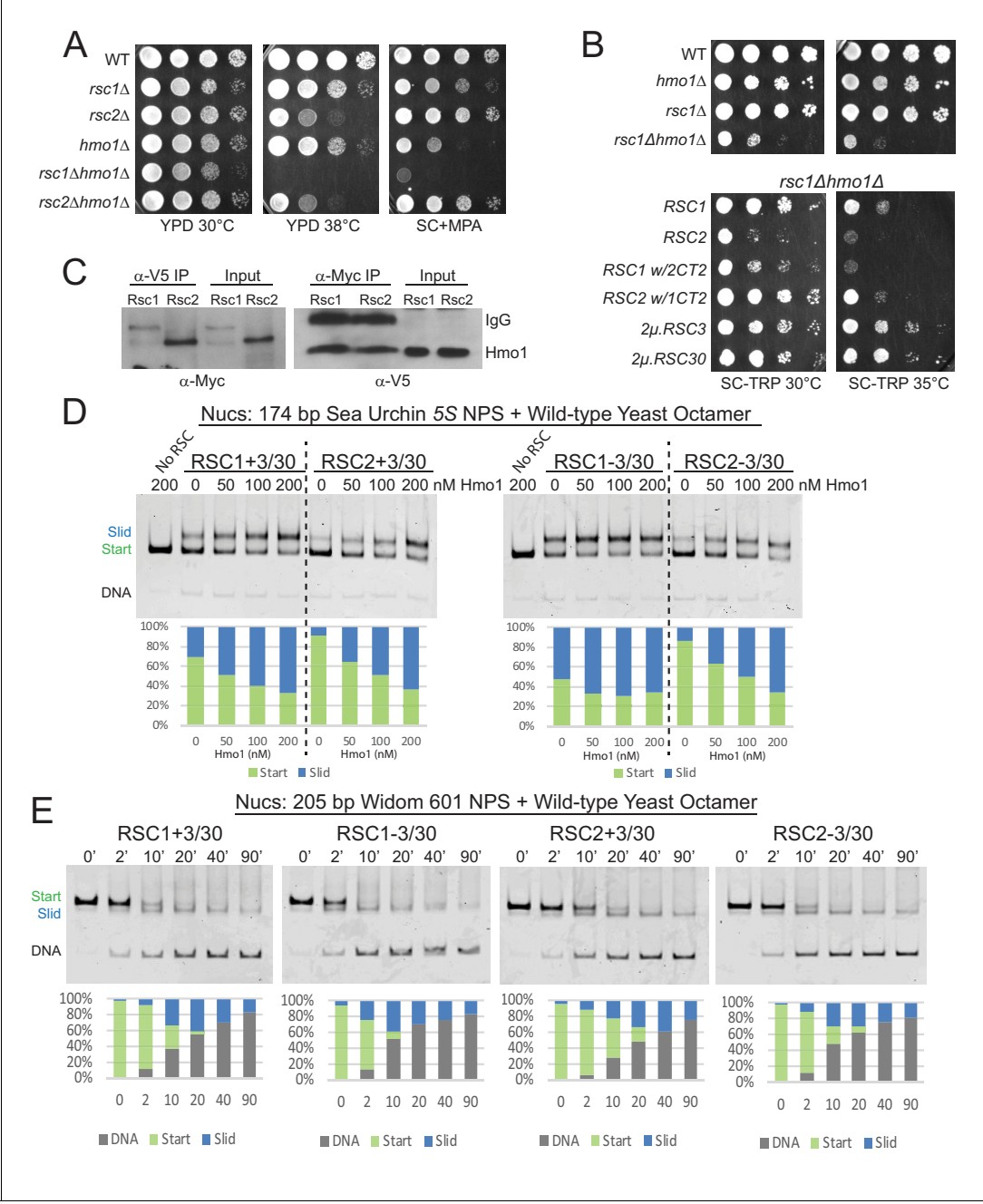

**Figure 4.** Hmo1 cooperates with RSC to remodel fragile or partially-unwrapped nucleosomes. (**A**) An *hmo1* null mutation is synthetically sick with *rsc1Δ*, but not *rsc2Δ*. WT (YBC604), *rsc1Δ* (YBC774), *rsc2Δ* (YBC82), *hmo1Δ* (YBC3509), *rsc1Δ* (YBC774), *rsc2Δ* (YBC82), *rsc1Δ hmo1Δ* (YBC3514), *rsc2Δ hmo1Δ* (YBC3515) spotted as 10x serial dilutions to YPD 30°C, YPD 38°C, and SC+20 μg/ml mycophenolic acid (MPA). One of two or more biological replicates shown. (**B**) The *rsc1Δ hmo1Δ* synthetic sickness is suppressed by high copy *RSC3*, *RSC30*, or *RSC1 CT2*. Strain *rsc1Δ hmo1Δ* (YBC3514) transformed with TRP1-marked *RSC1* (p609), *RSC2* (p604), *RSC1 w/2CT2* (p3097), *RSC2 w/1CT2* (p3098), 2μ.*RSC3* (p929), 2μ.*RSC30* (p911), or vector (pRS314) spotted as 10x serial dilutions to SC-TRP 30°C or SC-TRP 35°C. One of four biological replicates shown. (**C**) Co-IP of Rsc1 and Rsc2 with Hmo1. Sonicated chromatin extracts from *RSC1.9XMYC HMO1.V5* (YBC3558) and *RSC2.9XMYC HMO1.V5* (YBC3559) were immunoprecipitated using anti-Myc or anti-V5. Western blots were probed with anti-Myc or anti-V5 antibodies. One of three biological replicates shown. (**D**) Comparative sliding by RSC1 and RSC2 complexes (10 nM) of 174 bp sea urchin *5S* yeast mononucleosomes (20 nM) pre-incubated with increasing concentrations of Hmo1 protein. Reactions were conducted at 30°C for 20 min. The Start (green) and Slid (blue) bands were quantified and reported as percent of the total signal. The free DNA band was negligible and not quantified. (**E**) Comparative sliding and ejection of Widom 601 yeast mononucleosomes (20 nM) by RSC1 and RSC2 complexes (10 nM). The nucleosomal Start (green), Slid (blue), and free DNA (grey) bands were quantified and reported as a percent of the total signal. The online version of this article includes the following source data and figure supplement(s) for figure 4:

*Figure 4 continued on next page*

*Figure 4 continued*

**Source data 1.** Hmo1 enhances RSC sliding.
**Source data 2.** RSC slides 601 wt yNucs.
**Figure supplement 1.** Co-Immunoprecipitation of Rsc1 and Rsc2 with Hmo1 from MNase-treated chromatin extracts.
**Figure supplement 2.** Quantification of Hmo1 stimulation of RSC sliding 174 bp *5S* wild-type yeast mononucleosomes as conducted in *Figure 4D*.
**Figure supplement 2—source data 1.** Hmo1 stimulates 10 nM RSC sliding of *5S* yNucs.
**Figure supplement 3.** Additional sliding assays and quantification of 174 bp *5S* yeast mononucleosomes with RSC and Hmo1.
**Figure supplement 3—source data 1.** Additional Hmo1 stimulation of RSC *5S* sliding.
**Figure supplement 3—source data 2.** Quantification of additional Hmo1 stimulation of RSC *5S* sliding.
**Figure supplement 4.** Quantification of RSC sliding 205 bp Widom 601 wild-type yeast mononucleosomes as conducted in *Figure 4E*.
**Figure supplement 4—source data 1.** Quantification of RSC sliding 601.
**Figure supplement 5.** RSC1 and RSC2 slide Widom 601 H3 R40A mononucleosomes similarly.
**Figure supplement 5—source data 1.** RSC slides 601 R40A yNuc.
**Figure supplement 5—source data 2.** 205 bp Widom 601 nucleosome mapping.
**Figure supplement 6.** Quantification of RSC sliding 205 bp Widom 601 H3 R40A yeast mononucleosomes as conducted in *Figure 4—figure supplement 5A*.
**Figure supplement 6—source data 1.** Quantification of RSC sliding R40A 601 yNuc.

robust remodeling of 601 H3 R40A nucleosomes by RSC2 (*Figure 4—figure supplement 5A* and *Figure 4—figure supplement 6* for quantification), and (by nuclease susceptibility) found that these nucleosomes remain largely wrapped (*Figure 4—figure supplement 5B*). Our data support the model that while the H3 R40A mutation can result in an open and more loosely wrapped nucleosome conformation, this can be overcome by a strong DNA positioning sequence. In addition, the data indicate that the αN helix H3 R40A mutation itself does not impair remodeling by RSC2 unless that mutation confers partial unwrapping due to the underlying DNA sequence. Lastly, our mapping data confirm that *5S* nucleosomes produced with yeast octamers are less well-wrapped than 601 nucleosomes, and have DNA ends that are more susceptible to nuclease digestion.

## Rsc1 and Rsc2 both occupy wide NDRs, with Rsc1 specifically occupying tDNAs

RSC has been examined by chromatin co-immunoprecipitation (ChIP) in several formats (*Damelin et al., 2002*; *Yen et al., 2012*; *Vinayachandran et al., 2018*; *Brahma and Henikoff, 2019*), though only one study has examined Rsc1/2 differences. This early ChIP approach with microarrays examined Rsc1 and Rsc2 differences, and revealed similar promoter targets for RSC1 and RSC2 complexes (*Ng et al., 2002*). However, possible issues of sensitivity with earlier data combined with the ability to perform more advanced approaches and analyses prompted ChIP-seq experiments to reveal possible differences. Recent examination of Sth1/RSC occupancy by the 'cut and run' MNase approach (which would have superimposed and not distinguished between Rsc1 and Rsc2 occupancy) revealed general enrichment for RSC complexes at many +1 and −1 nucleosomes, as well as high enrichment of RSC within the NDR of genes with very wide NDRs (e.g., ribosomal protein genes). Notably, these wide NDR regions/promoters have been shown to contain RSC bound to partially-unwrapped nucleosomes, rather than the complete lack of nucleosomes (*Brahma and Henikoff, 2019*). Therefore, we will hereafter refer to them as 'partially-unwrapped regions' when appropriate, and use the term 'NDR' to generally refer to the nucleosome-deficient regions between the −1 and +one nucleosome.

To further examine the genome-wide location of RSC1 and RSC2 complexes and their relation to nucleosome wrapping, we utilized MYC-tagged derivatives of Rsc1, Rsc2 and Rsc3 (tagged at their endogenous loci) and performed MNase-based ChIP-seq from logarithmically growing cells. We observe RSC at locations similar to prior work (*Brahma and Henikoff, 2019*), and largely comparable profiles between Rsc1 and Rsc2 at most Pol II genes (as described previously *Ng et al., 2002*). However, we observe differential occupancy in two regions: promoters with wide NDRs and tDNAs.

First, we observed high occupancy of RSC (Rsc1, Rsc2 and Rsc3; *Figure 5A*) at promoters with wide NDRs, with Rsc1 appearing more enriched than Rsc2. As promoters with partially-unwrapped nucleosomes often contain the HMGB protein, Hmo1, we also compared the occupancy of Hmo1 using the ChIP-seq data (*Knight et al., 2014*) reprocessed using our parameters. Notably, Hmo1

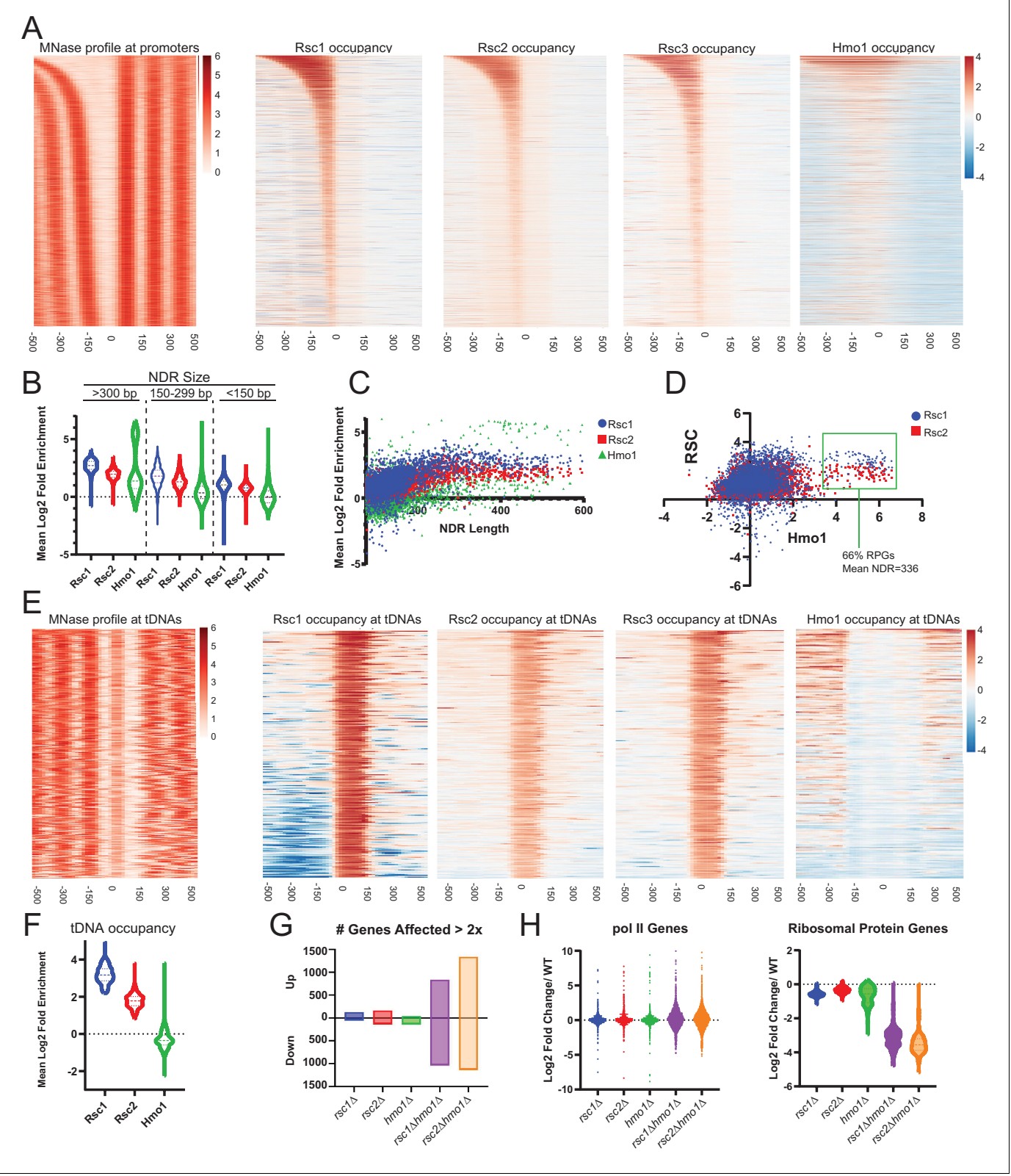

**Figure 5.** Rsc1 and Rsc2 occupy promoters with wide NDRS, with preferential occupancy of RSC1 at tDNAs. (**A**) Heat maps showing enrichment of nucleosomes, Rsc1, Rsc2, Rsc3, and Hmo1 at promoters, sorted by NDR length. (**B**) Violin plot of Rsc1, Rsc2, and Hmo1 occupancy at promoters at three categories of NDR length. (**C**) Plot of Rsc1, Rsc2, and Hmo1 occupancy compared to NDR length. (**D**) Plot of Rsc1 and Rsc2 enrichment by Hmo1 occupancy. (**E**) Heat maps showing enrichment of nucleosomes, Rsc1, Rsc2, Rsc3, and Hmo1 at all tDNAs (tRNA encoding genes). (**F**) Violin plot of Rsc1,

*Figure 5 continued on next page*

*Figure 5 continued*

Rsc2, and Hmo1 mean log2 fold enrichment at tDNAs. (G) Number of genes affected by rsc and hmo1 deletions. For each mutation the number of genes up or downregulated two fold or more compared to WT, *rsc1Δ* (↑129↓45), *rsc2Δ* (↑160↓129), *hmo1Δ* (↑44↓132), *rsc1Δ hmo1Δ* (↑838↓1028), *rsc2Δ hmo1Δ* (↑1336↓1131). (H) Gene Expression changes. Violin plots of RNA expression for each mutation at all pol II genes (6145 genes) and at ribosomal protein genes (132 genes) as compared to WT expression. ChIP-seq and RNA-seq data shown represents averages of two and three biological replicates, respectively.

The online version of this article includes the following source data and figure supplement(s) for figure 5:

**Source data 1.** Files for generation of occupancy heat maps.
**Source data 2.** RSC occupancy at promoters and tDNAs.
**Source data 3.** Differential gene expression.
**Figure supplement 1.** Gene expression changes in *rsc1Δ*, *rsc2Δ*, and *hmo1Δ* mutants.
**Figure supplement 1—source data 1.** List of ESR affected genes.
**Figure supplement 2.** Gene expression changes in *rsc1Δ*, *rsc2Δ*, and *rsc30Δ* mutants.
**Figure supplement 3.** Appearance of *RSC1/RSC2* and *RSC3/RSC30* in yeast evolution.

occupancy positively correlates with regions displaying the widest NDRs and bearing the highest Rsc1 and Rsc2 occupancy, with an apparent higher correlation with Rsc1 (*Figure 5A–D*). The loci with the highest Hmo1 occupancy displayed a mean region size of 336 bp, and a remarkable two-thirds of those genes were ribosomal protein genes (RPGs). Ribosomal protein gene promoters contain GC-rich sequences within their 'NDR' region, and in keeping, we observe Rsc3 enrichment, consistent with Rsc3/30 involvement in targeting or retention.

The most striking difference between Rsc1 and Rsc2 occupancy was observed at tDNAs. tDNAs encode tRNAs, and are approximately the size of a single nucleosome - though they are among the most nucleosome-depleted loci in the yeast genome, due at least in part to the action of RSC (*Parnell et al., 2008*) and their high fractional occupancy by RNA polymerase III transcription factors (*Kumar and Bhargava, 2013*). Here, Rsc1 complexes are markedly more enriched than Rsc2 complexes, and Hmo1 is notably absent (*Figure 5E,F*). Notably, many tDNAs are flanked by highly AT-rich sequences (*Giuliodori et al., 2003*), which could (in principle) confer partial unwrapping during the uncommon times (e.g., after replication) when tDNAs are transiently unoccupied by Pol III; here, RSC1 complexes might conduct nucleosome sliding to reveal tDNA sequences to the Pol III machinery. Indeed, the high fractional occupancy of Pol III complexes likely underlies the lack of Hmo1 at tDNAs, as Pol III factors would likely compete with Hmo1 for DNA binding and can remove nucleosomes during transcription. Taken together, Rsc1 and Rsc2 both highly occupy Pol II promoters known to contain partially-unwrapped nucleosomes and Hmo1, whereas Rsc1 preferentially occupies tDNAs.

## RSC and HMO1 cooperate to regulate gene expression

To address the transcriptional effects of each of these proteins, we performed RNAseq on logarithmically growing WT, *rsc1Δ*, *rsc2Δ*, *hmo1Δ*, *rsc1Δ hmo1Δ*, and *rsc2Δ hmo1Δ* cells cultured in SC media. We observe modest differences in transcriptional profiles between WT, *rsc1Δ*, and *rsc2Δ* (*Figure 5G–H*), which we interpret to be reflective of the known redundancy between Rsc1 and Rsc2 for most functions.

However, combining *rsc1Δ* or *rsc2Δ* with *hmo1Δ* resulted in a strong transcriptional shift, resulting in the upregulation (>2 fold) and downregulation (>2 fold) of ~1000 genes (*Figure 5G*), with the most affected class of genes including RPGs, which are downregulated (*Figure 5H*). Additionally the transcriptional shift observed in the double mutants largely mirrors the shift of both up-regulation and down-regulation observed in response to environmental stress response (ESR) genes (*Gasch et al., 2000*; *Brion et al., 2016*; *Figure 5—figure supplement 1*). These results suggest that Rsc1 and Rsc2 both cooperate with Hmo1 to promote the transcription of ribosomal protein genes, and that the burden for chromatin remodeling at these and other growth regulated loci requires the action of both Rsc1 and Rsc2, if Hmo1 is absent.

As Rsc3/30 also interacts with and affects Rsc1/2 function, we further analyzed *rsc1Δ rsc30Δ*, and *rsc2Δ rsc30Δ* mutants (*Figure 5—figure supplement 2*). Remarkably, only *rsc2Δ rsc30Δ* mutants

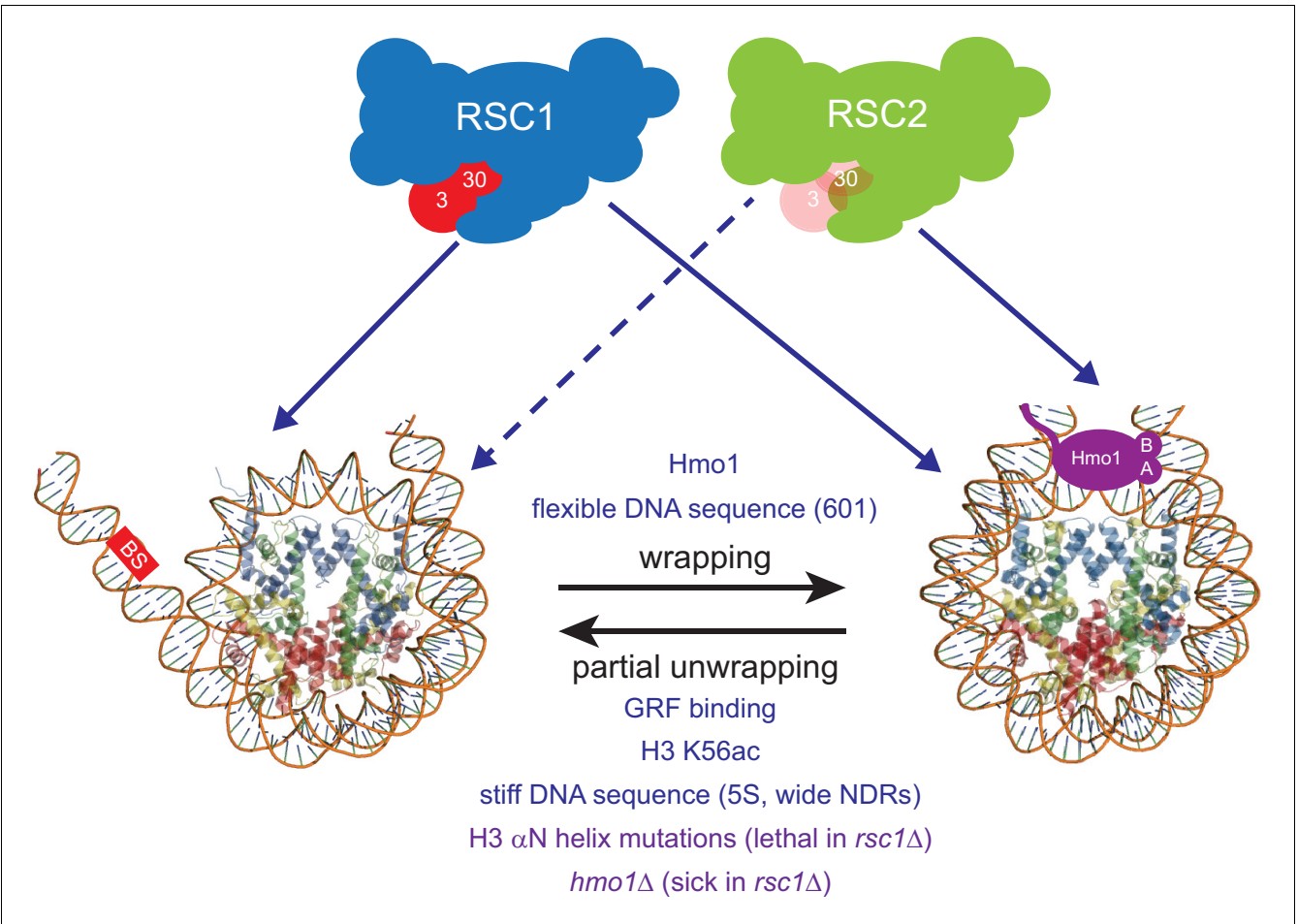

**Figure 6.** Model for RSC1 and RSC2 action on wrapped versus partially-unwrapped nucleosomes. Here, flexible/positioning DNA sequences and the protein Hmo1 promote wrapping (*Anderson et al., 2002*; *Iyer, 2012*; *Panday and Grove, 2017*). In contrast, partial unwrapping can be facilitated by 'stiff' (AT-rich) DNA sequences, acetylation (e.g., H3 K56ac), the binding of general regulatory transcription factors (not shown) to entry/exit DNA, or by mutation of residues within/near the H3 αN helix (which normally binds entry/exit DNA) (*Segal and Widom, 2009*; *Neumann et al., 2009*; *Knight et al., 2014*; *Ferreira et al., 2007*). Whereas fully-wrapped nucleosomes are remodeled well by RSC1 or RSC2 complexes, partially-unwrapped nucleosomes are better managed and remodeled by RSC1 complexes.

strongly impact RP genes – suggesting that the RSC1 complex, which interacts more strongly with Rsc3/30, is more reliant on Rsc3/30 than is the RSC2 complex, and that in a *rsc2Δ rsc30Δ* both complexes are impaired (*Figure 5—figure supplement 2C*).

Given our exploration of Rsc1/2 paralog and Rsc3/30 paralog function, we also explored their evolution. A whole genome duplication (WGD) occurred within the *Saccharomyces* lineage approximately 150 million years ago (*Wolfe and Shields, 1997*) resulting in *RSC1/2* paralogs. We find the single *RSC1/2* ortholog present in species that did not undergo the WGD (e.g., *Zygosaccharomyces*, *Ashbya*, and *Lachancea*), more closely related to *RSC2* than *RSC1*, suggesting *RSC2* as the more ancient ortholog (*Figure 5—figure supplement 3*). Notably, species more distant to *S. cerevisiae* lack both *RSC3* and *RSC30*. Following the appearance of *RSC3*, the WGD event then created *RSC1/2* and *RSC3/RSC30* orthologs, along with duplications of the ribosomal protein genes, a large fraction of which have been maintained in *Saccharomyces cerevisiae*. Finally, *HMO1* predates the appearance of *RSC3* and the WGD and is found in *Schizosaccharomyces pombe* (*Albert et al., 2013*). Thus, specialization of Rsc1 to preferentially bind Rsc3/30 and play a role in the regulation of the ribosomal protein genes was enabled by the WGD, and may have arisen to help rapidly and properly regulate RPGs in response to growth conditions.

Taken together, our results are consistent with RSC1 complexes bearing a higher intrinsic ability to mobilize partially-unwrapped nucleosomes compared to RSC2, augmented by partner proteins that preferentially assist either RSC1 (Rsc3/30) or RSC2 (Hmo1) complexes to more efficiently remodel at locations with partially-wrapped nucleosomes (*Figure 6*).

## Discussion

Genomes are diverse in DNA sequence composition, and the biophysical properties of DNA – in particular, DNA curvature and stiffness – creating a spectrum of affinities for nucleosomes, the main packaging unit of chromatin. This spectrum can undergo selection to create regions where nucleosome formation, positioning, and/or turnover is favored or disfavored, properties which can be utilized at promoters and enhancers to help regulate transcription factor binding, transcription, and ultimately fitness. These biophysical properties work in concert (and sometimes in opposition) with the action of chromatin remodelers, which utilize ATP to move nucleosomes to either favored or disfavored positions, and to eject nucleosomes to provide regulated access of transcription factors to DNA. The commonness in yeast of stiff/disfavored DNA sequences at proximal promoters, juxtaposed to bendable/favorable sequences at the −1 and +1 positions (especially at constitutive or highly transcribed genes), raises the possibility that chromatin remodelers may have undergone specialization to manage the remodeling of both fully wrapped and partially-unwrapped nucleosomes.

We begin by briefly discussing prior studies that address the extent of partial unwrapping of *5S* or 601 DNA sequences. It has been established that the 601 sequence displays higher overall affinity for the histone octamer than does the *5S* sequence (*Thåström et al., 1999*). However, a separate report found that the 601 DNA sequence (with *Xenopus* histones) displayed more unwrapping from the nucleosome edge (entry/exit) than does the *Xenopus 5S* sequence (*North et al., 2012*). It is worth noting, this work inserted a LexA binding sequence into both the *5S* and 601 sequences (by replacing the existing sequence) which involves the region that unwraps from the octamer. Notably, a set of subsequent studies (*Chen et al., 2014*; *Mauney et al., 2018*) used small angle X-ray scattering (SAXS) to compare the native/unaltered sea urchin *5S* to 601 nucleosomes, which showed that the *5S* sequence unwraps more rapidly and at lower salt concentrations than does 601.

Here, we provide several lines of evidence that the RSC1 complex can slide partially-unwrapped nucleosomes better than its paralog, the RSC2 complex. Furthermore, as RSC is assisted by additional proteins (e.g., Hmo1) that augment this function in vivo, RSC can be thought of as two functional entities: 'Rsc2 with Hmo1' and 'Rsc1 with Rsc3-30', with strong phenotypes only observed when both entities are impaired (*Figure 6*). First, we find that RSC1 acts more efficiently on *5S* nucleosomes, and retains activity on nucleosomes bearing H3 R40A, a mutation that favors unwrapping, whereas RSC2 is relatively impaired. Second, an unbiased genetic screen revealed synthetic lethality with *rsc1*Δ αN helix combinations, but not *rsc2*Δ αN helix combinations, strongly suggesting that the RSC2 complex has more difficulty remodeling partially-unwrapped nucleosomes. Third, Hmo1 stabilizes nucleosomes (*Panday and Grove, 2016*; *Panday and Grove, 2017*), and we observe synthetic phenotypes with *rsc1*Δ *hmo1*Δ mutants, but not with *rsc2*Δ *hmo1*Δ mutants (*Figure 4A*), suggesting that Rsc2 relies much more on Hmo1 for functional cooperativity than does Rsc1. This result is paralleled in our remodeling experiments, which show a more pronounced rescue of *5S* nucleosome sliding by RSC2 complexes than RSC1 complexes with recombinant Hmo1 (*Figure 4D*). We note that as nucleosome binding affinity is very similar between RSC1 and RSC2 complexes, the defect/challenge in remodeling with the RSC2 complex may involve a step downstream of binding, perhaps involving the ability of the complex to commit to the initiation of DNA translocation, which may be sensitive to the conformation of the nucleosome. Alternatively, nucleosome conformation/wrapping may impact the efficiency of DNA translocation – termed 'coupling' – which involves the probability of each ATP hydrolysis resulting in 1 bp of productive DNA translocation, a property known to be regulated in RSC complex by the actin-related proteins, Arp7 and Arp9 (*Clapier et al., 2016*; *Szerlong et al., 2008*). Finally, we emphasize that partially-unwrapped nucleosomes are not preferred by RSC1 complex over wrapped nucleosomes; RSC1 simply manages partially-unwrapped nucleosomes better than does RSC2 complex.

Initially, it may seem counter-intuitive for partial unwrapping of a nucleosome to impede rather than enhance RSC remodeling. However, we note that partial unwrapping is not an intermediate to full unwrapping by CRCs – as full unwrapping followed by full octamer re-wrapping is not the

mechanism of nucleosome sliding. Therefore, a partially-unwrapped nucleosome is not necessarily a remodeling intermediate, and therefore not necessarily stimulatory. Instead, we suggest that sliding involves a 1 bp/ATP DNA translocation mechanism imposed on a fully wrapped, or partially-wrapped nucleosome. Here, we suggest that partial unwrapping results in a nucleosome conformation that can indeed be encountered/bound by RSC1/2 complexes, but resists transition to the DNA translocation phase of remodeling – with RSC2 complexes less able to conduct this transition than RSC1 complexes (*Figure 6*). A second related model would be that RSC1 complex might better induce or stabilize a wrapped nucleosome conformation that is more conducive to initiating or continuing remodeling.

Above, we show that the product of remodeling (using *5S* nucleosomes formed with yeast octamers) migrates more slowly than does the initial substrate, and (by using Oregon Green-labeled H2A) that the product is not a hexasome or tetrasome involving H2A-H2B dimer loss. Nucleosome migration is determined by both the octamer position along the DNA (center vs end-positioned), and the overall shape of the octamer-DNA complex, including the extent to which the product/slid nucleosome is wrapped – as partial unwrapping and an 'open' conformation would be predicted to result in a slower migrating species through a native gel, just as a more 'compact' conformation results in a faster migrating species (*Chakravarthy et al., 2012*).

The mammalian polybromo protein (PBRM1) helps define the PBAF sub-complex of mammalian SWI/SNF complex, and is similar in domain composition to the combination of Rsc1/2 and the Rsc4 protein (which contains multiple bromodomains). PBRM1 also contains an HMG domain, which is notably absent in the Rsc1, 2, and 4 combination of domains. Here, we speculate that the Hmo1 protein and the HMG domain of PBRM1 may have functional similarities in managing DNA wrapping, which can be tested in future work.

RSC contains two proteins with affinity for GC-rich sequences: the paralogs Rsc3 and Rsc30. Here, we reveal a higher avidity of the Rsc3/30 module for the RSC1 complex, localize the region of interaction of Rsc3/30 with Rsc1/2 to the beginning of the CT2 domain, and provide genetic evidence that the preferred interaction of Rsc3/30 with Rsc1 has functional consequences. A current curiosity is the observation – true for both Rsc1 and Rsc2 complexes – that the presence of Rsc3/30 moderately inhibits remodeling, while the genetics supports a positive role for Rsc3/30 in assisting Rsc1/2 function. One obvious role for Rsc3/30 is in targeting RSC to GC-rich promoter sequences such as ribosomal protein genes, which likely underlies the essential nature of Rsc3/30 function. Here, future work may explore whether Rsc3/30 serve a regulatory role in the remodeling reaction, with their modest attenuation function relieved in the proper regulatory contexts of DNA sequence (e.g., GC richness) and protein composition (e.g., Hmo1, others) to help confer environmental sensing and properly regulate RPGs.

In favorable growth conditions, approximately 50% of the transcriptional effort of RNA Pol II is directed at RPGs (*Warner, 1999*). Our work supports and greatly extends the prior work of others (*Knight et al., 2014*; *Hepp et al., 2017*; *Wade et al., 2004*) that *S. cerevisiae* has evolved many ways to 'poise' chromatin at RPG promoters in a relatively 'open and ready' format. These promoters have evolved to be less favorable to nucleosomes by bearing regions of 'stiff' AT-rich DNA (*Segal and Widom, 2009*) (note: Rsc1/2 each bear an AT hook), are punctuated by GC-rich DNA (known Rsc3/30 binding sites), and use histone acetylation (e.g., H3 K56ac) (*Neumann et al., 2009*), Hmo1 (*Hall et al., 2006*), DNA-binding general regulatory factors (e.g., Rap1, Abf1) and chromatin remodeling complexes such as RSC and SWI/SNF complex – that together help keep these regions nucleosome deficient and partially-unwrapped (*Kubik et al., 2018*; *Hepp et al., 2017*; *Brahma and Henikoff, 2019*; *Reja et al., 2015*). While RSC1 and RSC2 complexes appear largely redundant at most genes, and can both be assisted at RPG promoters by Hmo1, our work supports the notion that the RSC1 complex, through its ability to better manage partially-unwrapped nucleosomes and preferential association with Rsc3/30, has become specialized to help perform this role (*Figure 6*). This combination of specialization and partial redundancy provides this system the needed robustness and the ability to conduct rapid and sophisticated activation/regulation of this important RPG class – attributes which may contribute to fitness in diverse environments.

# Materials and methods

## Key resources table

| Reagent type (species) or resource | Designation | Source or reference | Identifiers | Additional information |
|---|---|---|---|---|
| Gene *Saccharomyces cerevisiae* | Yeast Genome | UCSC | SacCer3 | |
| Strain, strain background (*Escherichia coli*) | BL21-CodonPlus (DE3)RIL | Agilent | Cat# 230245 | |
| Antibody | Anti-Sth1 (Rabbit polyclonal) | *Cairns et al., 1996* | | (1:1000) |
| Antibody | Anti-Rsc3 (Rabbit polyclonal) | *Angus-Hill et al., 2001* | | (1:1000) |
| Antibody | Anti-Rsc30 (Rabbit polyclonal) | *Angus-Hill et al., 2001* | | (1:1000) |
| Antibody | Anti-Rsc2 (Rabbit polyclonal) | *Kasten et al., 2004* | | (1:1000) |
| Antibody | Anti-Rsc4 (Rabbit polyclonal) | *Kasten et al., 2004* | | (1:1000) |
| Antibody | Anti-HA (mouse monoclonal) | *Cairns et al., 1999* | | (1:1000) |
| Antibody | Anti-Myc (mouse monoclonal) | Abcam | Cat# ab56 RRID:AB_304876 | (1:1000) |
| Antibody | Anti-V5 (mouse monoclonal) | Thermo Scientific | Cat# R960-25 RRID:AB_2556564 | (1:1000) |
| Peptide, recombinant protein | 8XHIS.HMO1 | This paper, *Figure 4* | | purified from *E. coli* BL21-Codon Plus(DE3)-RIL cells |
| Peptide, recombinant protein | Histone H2A, Oregon green, yeast octamers | *Xin et al., 2009* | | |
| Peptide, recombinant protein | Histone H3 R40A | This paper, *Figure 3* | | purified from *E. coli* BL21-Codon Plus(DE3)-RIL cells |
| Peptide, recombinant protein | AcTEV protease | Thermo Scientific | Cat# 12575015 | |
| Peptide, recombinant protein | Micrococcal Nuclease | USB | Cat# 70196Y | |
| Peptide, recombinant protein | Exonuclease III | New England Biolabs | Cat# M0206 | |
| Peptide, recombinant protein | S1 nuclease | Thermo Scientific | Cat# 18001–016 | |
| Peptide, recombinant protein | Klenow fragment | New England Biolabs | Cat# M0212L | |
| Commercial assay or kit | NEBNext ChIP-Seq MasterMix Set | New England Biolabs | Cat# E6240L | |
| Commercial assay or kit | RNEasy | Qiagen | Cat# 74106 | |
| Commercial assay or kit | RiboPure RNA purification kit, Yeast | Thermo Scientific | Cat# AM1926 | |
| Commercial assay or kit | TruSeq Stranded Total RNA Library Prep Kit with Ribo Zero Gold | Illumina | Cat# RS-122–2301 | |

*Continued on next page*

*Continued*

| Reagent type (species) or resource | Designation | Source or reference | Identifiers | Additional information |
|---|---|---|---|---|
| Commercial assay or kit | BioRad protein assay reagent | BioRad | Cat# 500–0006 | |
| Commercial assay or kit | Minelute PCR purification kit | Qiagen | Cat#28006 | |
| Chemical compound, drug | 5-Fluoroorotic Acid (5FOA) | Toronto Research Chemicals | Cat# F59500 | |
| Chemical compound, drug | Mycophenolic Acid (MPA) | Calbiochem | Cat# 475913 | |
| Software, algorithm | Prism | GraphPad Software Inc | RRID:SCR_002798 | Version 8.0.2 |
| Software, algorithm | DESeq2 | Bioconductor | RRID:SCR_015687 | Version 1.21 |
| Software, algorithm | STAR | *Dobin et al., 2013* | RRID:SCR_015899 | Version 2.5.4 |
| Software, algorithm | R | *R Development Core Team, 2018* | RRID:SCR_001905 | |
| Software, algorithm | pHeatmap | *Kolde, 2019* | RRID:SCR_016418 | |
| Software, algorithm | Novoalign | Novocraft | RRID:SCR_014818 | Version 3.8.2 |
| Software, algorithm | UMI Scripts | Huntsman Cancer Institute | See Materials and methods for github link | |
| Software, algorithm | BioToolBox packages | TJ Parnell, Huntsman Cancer Institute | See Materials and methods for github link | |
| Software, algorithm | ImageQuant TL | GE Healthcare | RRID:SCR_018374 | |
| Other | IgG Sepharose Fast Flow | GE Healthcare | Cat# GE 17-0969-01 | |
| Other | Calmodulin Affinity Resin | Agilent | Cat# 214303 | |
| Other | Nickel-NTA Agarose Beads | Qiagen | Cat# 30230 | |
| Other | Slide-A-Lyzer mini dialysis units | Thermo Scientific | Cat# 69560 | |
| Other | Dynabeads Pan Mouse IgG | Thermo Scientific | Cat#11041 | |

## Strains, Media, Yeast growth, and Assay Replication

Resources used in this study are provided in the Key Resources Table. Rich media (YPD), synthetic complete (SC), and sporulation media were prepared using standard methods. Standard procedures were used for transformations, sporulation, tetrad analysis and spotting. A null mutation of Hmo1 was obtained from Invitrogen and crossed into *rsc1* and *rsc2* deletion strains. Rsc1 was C-terminally TAP-tagged as described by *Rigaut et al., 1999*, and Hmo1 C-terminally tagged with V5 as described in *Funakoshi and Hochstrasser, 2009*. Full genotypes of yeast strains are available in *Supplementary file 1*. A technical replicate is defined as a test of the same sample multiple times, whereas a biological replicate is defined as the same test run on multiple biological samples of independent origin. The number and type of replicates for each experiment is indicated in the figure legend.

## Plasmid construction

DNA encoding Rsc1 w/Rsc2 CT2 was created by PCR amplification of RSC2 CT2 from position K617 through the end of the gene, using primers containing 40 bp homology to *RSC1* or vector sequence. PCR product was co-transformed with linearized p609 (314.*RSC1*) into YBC800 (*rsc1Δ rsc2Δ* [316. *RSC1*]) and homologous recombination inserted the DNA encoding CT2 into *RSC1* beginning at position K617. Rsc2 w/1CT2 at position K656 was created by amplifying *RSC1* CT2 on p609 (314.

*RSC1*) and co-transforming into linearized p604 (314.*RSC2*) into YBC803 (*rsc1Δ rsc2Δ* [316.*RSC2*]) and repaired by homologous recombination. Plasmid was isolated from TRP positive colonies, sequenced, and retransformed to check for complementation. Additional swaps were constructed similarly. Internal deletions in Rsc1 and Rsc2 were created by digesting with restriction enzymes with blunt or compatible ends, removing the small fragment and ligating ends back together. Plasmid markers were swapped as needed as described in *Cross, 1997*. A full list of plasmids used in this study is provided in *Supplementary file 2*.

## RSC1 and RSC2 TAP purification

RSC complexes were purified from *S. cerevisiae* strains BCY211 (RSC2-TAP) and BCY516 (RSC1-TAP) harboring an integrated version of the C-terminally Tandem Affinity Purification (TAP)-tagged on the construct of the indicated gene. Purifications were performed essentially as described (*Rigaut et al., 1999*; *Szerlong et al., 2008*) with the following modifications. After harvesting, cells were washed with 0.5x PBS + 10% glycerol, pelleted, and frozen in liquid nitrogen as pea-size pieces. Frozen cells were pulverized in a SPEX SamplePrep 6850 Freezer/Mill at a rate of 10, for 10 cycles of 3 min 'on' and 2 min 'off', with the resulting powder stored at −80°C until purification.

RSC1+3/30 complex was purified from BCY516 essentially as described (*Szerlong et al., 2008*) with the following modifications since Rsc1 is found at lower levels than Rsc2 in yeast cells. Pulverized cells (160 g) were solubilized in Lysis Buffer (50 mM HEPES [pH 7.5], 400 mM KOAc, 10 mM EDTA, 20% glycerol, 0.5 mM BME, protease inhibitors). The KOAc concentration of the lysate was brought to 450 mM and nucleic acids were precipitated with 0.1–0.2% polyethyleneimine. The cleared lysate was incubated with 5 ml IgG Sepharose 6 Fast Flow (GE Healthcare) for 3 hr, the resin sequentially washed with Lysis Buffer, IgG Wash Buffer (20 mM Tris-Cl, [pH 8], 400 mM KOAc, 0.1% Igepal, 1 mM EDTA, 0.5 mM BME, protease inhibitors), and IgG Elution Buffer (20 mM Tris-Cl [pH 8], 200 mM KOAc, 10% glycerol, 0.1% Igepal, 0.5 mM BME). The RSC1-TAP complex was cleaved from the IgG resin with 500 U AcTEV enzyme (Invitrogen) gently rotating the slurry at 15°C for 1 hr followed by 4°C rotation overnight. The eluate was collected, brought to 3 mM $CaCl_2$, and bound to 1.5 ml Calmodulin Affinity Resin (Agilent Technologies) overnight. The resin was washed with 250 Calmodulin Wash Buffer (20 mM Tris-Cl [pH 7.5], 250 mM NaCl, 1 mM Mg Acetate, 1 mM imidazole, 2 mM CaCl2, 10% glycerol, 0.1% Igepal, 0.5 mM BME, protease inhibitors). The RSC1 complex was eluted with 250 Calmodulin Elution Buffer (20 mM Tris-Cl [pH 7.5], 250 mM NaCl, 2 mM EGTA, 10% glycerol, 0.1% Igepal, 0.5 mM BME, protease inhibitors) in successive 500–750 µl fractions. The RSC1-containing fractions were determined, pooled, and residual nucleic acid was removed by a final DEAE Sepharose clean-up step. The complex was concentrated on a Vivaspin 30K PES concentrator, concentration was determined by Bradford assay (Bio-Rad) and confirmed by silver stained polyacrylamide SDS-PAGE analysis.

RSC1-3/30 complex was purified as described above except the IgG Wash Buffer had 750 mM NaCl instead of KOAc, and the IgG resin was rotated overnight during the IgG Wash step to remove the Rsc3/30 module. The bound IgG resin was equilibrated into IgG Elution Buffer before cleaving with AcTEV.

RSC2+3/30 complex was purified as above with the following modifications. Approximately 80 g of pulverized BCY211 cells were solubilized in Lysis Buffer. Cleared lysates were incubated with IgG Sepharose, washed and eluted with AcTEV enzyme as above. The eluate was collected, brought to 3 mM $CaCl_2$, and bound to 1.5 ml Calmodulin Affinity Resin for 3 hr. The resin was washed with 200 KOAc Calmodulin Wash (same as Calmodulin Wash but with 200 mM KOAc replacing the NaCl) followed by 150 NaCl Calmodulin Wash (same as Calmodulin Wash but with 150 mM NaCl). The RSC2 complex was eluted with 150 Calmodulin Elution Buffer (same as above but with 150 mM NaCl) in successive 500–750 µl fractions. The RSC2-containing fractions were determined and pooled. The salt was increased to 250 mM NaCl and residual nucleic acid was removed on DEAE Sepharose. The complex was immediately diluted to 125 mM NaCl while maintaining the other buffer component concentrations. The complex was concentrated and the concentration was determined as above.

RSC2-3/30 complex was purified essentially as the RSC2+3/30 complex with the following modifications. After binding the Calmodulin Affinity Resin, the resin was washed with 200 KOAc Calmodulin Wash Buffer, followed by four 20 min washes with 500 mM NaCl Calmodulin Wash Buffer, and a final rinse with 150 NaCl Calmodulin Wash Buffer to equilibrate the resin. The RSC2-3/30 complex

was eluted from the Calmodulin Affinity Resin, residual nucleic acid removed on DEAE Sepharose, concentrated and quantified as above for the RSC2+3/30 complex.

## Hmo1 purification

Recombinant Hmo1 was purified from bacteria as an N-terminal fusion to 8xHIS (plasmid from T. Formosa). Briefly, *E. coli* BL21-CodonPlus(DE3)RIL transformed with pET.8xHIS.Hmo1 was induced with IPTG at 30°C for 3 hr to express the fusion protein. Cells were harvested, resuspended in HIS Lysis Buffer (20 mM Tris [pH 8], 500 mM NaCl, 10 mM imidazole, 10% glycerol, 0.1% Igepal, 1.5 mM BME, protease inhibitors), and lysed by sonication. The lysate was cleared and bound to pre-equilibrated Ni-NTA agarose beads (Qiagen) for 1.5 hr. The bound resin was washed sequentially with HIS tag Lysis Buffer, Ni Wash Buffer A (same as HIS Lysis Buffer but with 30 mM imidazole), and Ni Wash Buffer B (same as HIS Lysis Buffer but with 50 mM imidazole). The protein was eluted with Ni Elution Buffer (20 mM Tris [pH 8], 500 mM NaCl, 250 mM imidazole, 10% glycerol, 0.005% Igepal, 1.5 mM BME, protease inhibitors) and sized on a Superdex 200 gel filtration column (GE). Hmo1 ran as a soluble aggregate (approximately 440 kD) larger than predicted from its molecular weight (27.5 kD). The Hmo1 protein was dialyzed into 150 mM Sizing Buffer (20 mM Tris [pH 8], 150 mM NaCl, 10% glycerol, 1.5 mM BME, protease inhibitors) and concentrated before storing at −80°C in aliquots.

## RSC1 and RSC2 Rsc3/30 Stringency Testing

Stringency testing on Rsc3/30 association with RSC1 and RSC2 was conducted on IgG Sepharose. Cells were grown and pulverized as for RSC purification. Sixty grams of RSC1-TAP cells and 20 g of RSC2-TAP cells were solubilized in Lysis Buffer and the lysate was cleared as above. One milliliter of IgG resin was incubated with the cleared lysate for 2 hr. The beads were washed with Lysis Buffer, and then with IgG Wash Buffer, before being split into 10 separate Eppendorf tubes for testing. The beads were gently pelleted, rinsed three times with the specific IgG Wash Buffer containing 150, 250, or 500 mM NaCl, as indicated. The specific IgG Wash Buffer was added for 1 hr or 16 hr (overnight), as indicated. All beads were washed three times with IgG Elution Buffer, resuspended in IgG Elution Buffer, and RSC was released from the beads with AcTEV enzyme as above and 1/10[th] of the eluate was analyzed by silver stain on a 6% SDS polyacrylamide gel.

## Co-Immunoprecipitations

Whole cell extracts were prepared and co-immunoprecipitations for RSC were performed as described, (*Cairns et al., 1999*) with the following modification. Anti-Myc or anti-HA antibodies (0.6 µg) were bound to a 25 µl slurry of pan-mouse magnetic Dynabeads (Thermo Fisher) and incubated with 1000 µg of extract. Blots were probed with anti-Sth1 (*Saha et al., 2002*), anti-Rsc3, anti-Rsc30 (*Angus-Hill et al., 2001*), anti-Rsc2 and anti-Rsc4 (*Kasten et al., 2004*).

Extracts for Rsc1 and Rsc2 chromatin co-IPs with Hmo1 were prepared from cells grown to $OD_{600} = 0.8$, Cultures were crosslinked in 1% formaldehyde final for 30 min at RT, quenched with 0.2M glycine, and lysed by bead-beating (mini-beadbeater, Biospec) in LB140 buffer. Samples were sonicated in a Bioruptor Pico (Diagenode) to release chromatin or MNase treated with 100U of micrococcal nuclease for 15 min at 37°C. Immunoprecipitations were performed with 1000 µg of extract and a 25 µl slurry of pan-mouse Dynabeads bound to anti-Myc (Abcam) or anti-V5 (Thermo-Fisher) antibodies.

## Nucleosome assembly

Yeast octamers were produced using purified *S. cerevisiae* histones expressed in *E. coli* BL21-CodonPlus(DE3)RIL. Either wild-type histone H3 or the H3 R40A mutant was assembled into yeast octamers by salt-dialysis, essentially as described previously (*Dyer et al., 2004*). Mononucleosomes were assembled from a linear salt gradient dialysis (2 M to 50 mM KCl) using Slide-A-Lyser Mini Dialysis units with a 7000 molecular weight cutoff (Thermo Scientific) essentially as described (*Clapier et al., 2016*) using one of several DNA positioning sequences. The 205 bp Widom 601 positioning sequence (*Lowary and Widom, 1998*) were produced from pUC12 × 601 digested with AvaI and purified using preparative electrophoresis (PrepCell, Bio-Rad). The 174 bp sea urchin *5S* positioning sequence was prepared by PCR amplification using a plasmid containing a copy of the

sea urchin *5S* sequence as a template. The PCR product was precipitated before purifying on a Pre-pCell. The exact 601 and *5S* positioning sequences used are given in *Supplementary file 2*. Assembled mononucleosomes were separated from free DNA by gradient sedimentation on a 10–30% sucrose gradient as previously described (*Wittmeyer et al., 2004*). Yeast octamers bearing H2A fluorescently labeled with Oregon Green on Q114C were a gift from L. McCullough and T. Formosa (*Xin et al., 2009*). These labeled yeast octamers were assembled into nucleosomes with the 174 bp *5S* NPS and purified as described above.

## ATPase assays

Measurement of ATP hydrolysis was as described previously (*Saha et al., 2002*; *Wittmeyer et al., 2004*) using a color (malachite green) absorbance assay that quantitatively measures released free phosphate. Time courses were performed on two separate purifications of each type of RSC complex. Reactions were performed in triplicate. Activities were measured at $V_{max}$ using double-stranded Bluescript plasmid as the substrate.

## Nucleosome sliding assay

Nucleosome sliding assays were performed as described (*Clapier et al., 2016*) with the following modifications. Reactions were conducted in 10 mM Tris [pH 7.4], 50 mM KCl, 3 mM $MgCl_2$, 0.1 mg/ml BSA, 1 mM ATP with 20 nM mononucleosomes at 30°C and 400 rpm shaking in a Thermomixer (Eppendorf). The amount of remodeler varied. Sliding reactions on 205 bp 601 nucleosomes used 10 nM RSC. Sliding reactions on 174 bp *5S* nucleosomes used 30 nM RSC unless otherwise indicated. Aliquots were removed at each time point and reactions were stopped by adding 10 mM EDTA + 200 ng competitor DNA (Bluescript plasmid). Samples were loaded with 10% glycerol on a 4.5% (37.5:1) native polyacrylamide gel and run in 0.4x TBE for 45 min at 110 V constant. Gels were stained with ethidium bromide solution and scanned on a Typhoon Trio (Amersham, GE).

## Nucleosome gel shift assays

Nucleosome gel shift assays were conducted similarly to the nucleosome sliding assay described above with the following modifications. Reactions were conducted in the absence of ATP and at 30°C for 20 min with 20 nM mononucleosomes. The RSC remodelers were added at 30 and 60 nM. The reactions were loaded on to 3.8% (37.5:1) native polyacrylamide gels and run in 0.4x TBE for 55 min at 110 V constant without addition of any stop solution or competitor. Gels were stained with ethidium bromide solution and scanned on a Typhoon Trio.

## Histone mutant screen

Null mutations in *rsc1* and r*sc2* were crossed into the H3/H4 shuffle strain WZY42 (*Zhang et al., 1998*). Strains YBC1939 (WZY42), YBC2090 and YBC3040 and YBC3221 and YBC3547 were transformed in 96 well plate format with a TRP-marked histone H3-H4 or H2A-H2B mutant plasmid library respectively, obtained from and screened as described previously (*Nakanishi et al., 2008*). Transformants were spotted to SC-TRP plates, replica plated again to SC-TRP plates after 2 days, followed by replica plating to SC-TRP and SC-TRP + 5FOA. Synthetic lethality with mutations in the H3 αN helix were confirmed by a second round of individual transformations and shuffles.

## Recombinant nucleosome mapping

The position and wrapping of the recombinant nucleosomes were determined by sequencing the protected DNA fragment after treating assembled nucleosomes with Exonuclease III and S1 nuclease. Approximately 400–800 fmol of nucleosomes purified from sucrose gradients was digested with a 5–25U titration of ExoIII enzyme (New England Biolabs) for 1, 2, or 3 min at 37°C in ExoIII Buffer (10 mM Tris [pH 8], 50 mM NaCl, 3 mM $MgCl_2$). Reactions were moved to ice, S1 Buffer and NaCl were added (30 mM NaOAc [pH 4.6], 1 mM ZnOAc, 5% glycerol, and 300 mM NaCl final concentration), and treated with 50U S1 for 30 min at room temperature. Tris [pH 8.8] and EDTA were added to final concentrations of 88 mM and 14 mM, respectively, and heated to 70°C for 10 min. SDS was added to 1%, vortexed, and iced. The protected DNA fragments were cleaned up on a Qiagen MinElute PCR purification column and eluted in 30 µl EB. The level of digestion was determined for each sample on a 4.5% (37.5:1) native polyacrylamide 0.4x TBE gel as described above.

Libraries were made from the above protected fragments using the NEBNext ChIP-Seq Master-Mix Set (New England Biolabs) with the following modifications. Samples did not go through the initial End-Repair. A custom adaptor with an 8 bp unique molecular identifier (UMI) was ligated onto the dA-tailed samples. The UMI Adaptor was created from an oligo based on the standard NEBNext adaptor sequence incorporating eight random nucleotides at the 5' end of the oligo (see oligo sequence in File Supplement 2). To create the UMI Adaptor, 25 µM oligo (synthesized by IDT) was first heated to 95°C and slow cooled to room temperature in Duplex Buffer (100 mM KOAc, 30 mM HEPES [pH 7.5]). The ends of the annealed UMI Adaptor were filled in using Klenow (New England Biolabs). The reaction was stopped with EDTA and heat. The adaptor was cleaned up on a Micro Bio-Spin six chromatography column (Bio-Rad) equilibrated with water. A dT-tail was added to the adaptor with Klenow exo- (New England Biolabs). The UMI Adaptor was cleaned up on a Micro Bio-Spin six column and eluted in water. The UMI Adaptor was diluted to 1.14 µM final concentration for use in library preparations. High-throughput sequencing was performed by Illumina's protocol for 50 bp paired-end runs on an Illumina HiSeq 2500 or a MiSeq.

The embedded UMI code was first extracted from the Fastq files using the script embedded_UMI_extractor from the package UMIScripts (https://github.com/HuntsmanCancerInstitute/UMI-Scripts; *Parnell, 2019a*). Output Fastq sequences were then aligned to an index comprised of the recombinant sea urchin *5S* or 601 sequence using Novocraft Novoalign (version 3.8.2), giving the adapter sequences for trimming. After alignment, PCR-duplicate reads were identified and marked based on the UMI information with the UMISripts application bam_umi_dedup. The 5' start positions for each alignment, discarding duplicates, were recorded as a bigWig file with 1 bp resolution using the application bam2wig from the BioToolBox package (https://github.com/tjparnell/biotoolbox; *Parnell, 2020a*). Separate bigWig files were generated for each length of alignments (92–169 bp). To normalize for sequencing read depth, alignment counts were scaled to an equivalent of 100K reads (calculated from the total sum of alignments without regard to alignment length). Count matrices for each length at each position on the reference sequence and for each sample were then collected using the BioToolBox application get_datasets from the bigWig files. Count matrices were analyzed with Microsoft Excel employing a 5% cutoff of the normalized fragment count.

## ChIP seq and analysis

Yeast cultures from yHN1 (YBC1544-RSC1.9XMYC) and yHN2 (YBC1545; RSC2.9XMYC) (*Ng et al., 2002*) were grown in SC-TRP at 30°C with two biological replicates. Cultures were harvested at $OD_{600}$ = 0.8, and crosslinked with 1% formaldehyde final for 30 min at room temperature. Cultures were quenched with 0.2M glycine for 5 min. Chromatin extracts were prepared by bead-beating, and chromatin was liberated with micrococcal nuclease. Immunoprecipitation was performed using anti-cMyc 9E11 (Abcam). DNA was isolated for input and IP samples and assembled into a library using Illumina protocols and sequenced as single end reads on an Illumina sequencer.

Fastq sequences were aligned to the yeast genome (UCSC version SacCer3) with NovoCraft Novoalign (version 3.8.2), allowing for one random alignment for multi-mapping reads. To maintain processing consistency between single-end and paired-end alignments, paired-end was aligned as single-end by ignoring the second read. Alignments were processed using the MultiRepMacsChIP-Seq pipeline (version 10.1, https://github.com/HuntsmanCancerInstitute/MultiRepMacsChIPSeq; *Parnell, 2020b*). Since MNase-digested chromatin yields high levels of coordinate-duplicate alignments (observed mean of 65%, range 57% to 74%), duplicate alignments were randomly subsampled to a uniform rate of 40% to remove sample bias while retaining relative signal intensity. Alignments over ribosomal DNA, telomeric sequences, mitochondrial chromosome, and other high copy sequences were excluded. Fragment coverage tracks were generated by extending alignments in the 3' direction by 160 bp in all fragments. We note that there may be some between-sample biases in the distribution of MNase-digested fragment lengths, particularly with RSC-enriched fragments, but rationalized that uniformity of processing should help minimize these biases. Replicates were depth-normalized (Reads Per Million) prior to combining as an average after confirming reasonable similarity to each through standard correlation metrics. Log2 fold enrichment of ChIP signal over non-enriched nucleosome signal (Input) was generated with Macs2 (*Zhang et al., 2008*) without background lambda correction. Hmo1 ChIPs (GSM1509041) (*Knight et al., 2014*) were re-processed in a similar manner.

NDRs were determined by first determining all nucleosome positions in the genome using the software package biotoolbox-nucleosome (https://github.com/tjparnell/biotoolbox-nucleosome; *Parnell, 2019b*) with the Input MNase sample alignments. To reduce mapping noise and increase efficacy of calling positions, 'skinny' nucleosome coverage was used rather than midpoint data. Fragment coverage was generated by shifting the 5' alignment coordinate in the 3' direction by 37 bp and extending 74 bp, essentially recording the predicted central portion of the nucleosome. After mapping nucleosomes, NDRs were extracted by calculating all inter-nucleosomal intervals and selecting for those with lengths between 75 and 600 bp and occurring over or adjacent to a defined protein-coding gene TSS. NDRs were sorted by decreasing length and aligned by the edge closest to the TSS; NDRs between divergent promoter pairs are represented once.

Data were collected with applications from the BioToolBox package (https://github.com/tjparnell/biotoolbox; *Parnell, 2020a*). Mean occupancy of RSC and Hmo1 were collected over the NDRs using the application get_datasets, while spatial data surrounding NDRs and tDNAs were collected with get_relative_data. Heat maps were generated in R (*R Development Core Team, 2018*) with pHeatmap (*Kolde, 2019*). Dot plots and violin plots were generated with GraphPad Prism.

## RNA seq and analysis

Yeast cultures for RNASeq were grown in SD media supplemented for auxotrophic amino acids at 30°C in three biological replicates. RNA from logarithmically growing cells was purified using Ambion Ribo-pure yeast kit. Samples were additionally DNAse-treated and cleaned up with Qiagen RNeasy kit. Illumina Ribo Zero yeast kit was used for library preparation, and sequencing performed on an Illumina sequencer.

Fastq reads were processed following an RNASeq pipeline (https://github.com/HuntsmanCancer-Institute/hciR; *Stubben, 2020*). Briefly, reads were aligned with STAR (version 2.5.4, *Dobin et al., 2013*, counts obtained with Subread featureCounts (version 1.6.3) based on Ensembl annotation (release 90), and differential gene expression performed with DESeq2 (version 1.21). Dot plots and violin plots were generated with GraphPad Prism (version 8.0.2).

## Acknowledgements

We thank Brian Dalley and the High-Throughput Sequencing Shared Resource for sequencing data. We thank Ali Shilatifard, Tim Formosa, Jef Boeke, Kevin Struhl, Anne Grove, Roger Kornberg and Sharon Dent for strains, plasmids, and labeled yeast octamer. We thank Patrick Murphy and Edward Grow for helpful discussions on data analysis and experimental design and Naveen Verma, Tim Formosa, and Mahesh Chandrasekharan for feedback on the manuscript. We thank HHMI for support of AS, MK and BC, and thank NCI CA042014 to Huntsman Cancer Institute for support of core facilities.

## Additional information

### Funding

| Funder | Grant reference number | Author |
|---|---|---|
| Howard Hughes Medical Institute | Brad Cairns Investigator | Alisha Schlichter Margaret M Kasten Bradley R Cairns |
| National Cancer Institute | CA042014 | Timothy J Parnell |

The funders had no role in study design, data collection and interpretation, or the decision to submit the work for publication.

### Author contributions

Alisha Schlichter, Margaret M Kasten, Conceptualization, Data curation, Formal analysis, Investigation, Methodology, Writing - original draft, Writing - review and editing; Timothy J Parnell, Data curation, Formal analysis, Methodology, Writing - original draft; Bradley R Cairns, Conceptualization, Supervision, Funding acquisition, Writing - original draft, Writing - review and editing

## Author ORCIDs

Alisha Schlichter  https://orcid.org/0000-0003-1928-9396
Margaret M Kasten  https://orcid.org/0000-0001-5753-7122
Timothy J Parnell  http://orcid.org/0000-0002-3632-3691
Bradley R Cairns  https://orcid.org/0000-0002-9864-8811

## Decision letter and Author response

Decision letter https://doi.org/10.7554/eLife.58130.sa1
Author response https://doi.org/10.7554/eLife.58130.sa2

## Additional files

### Supplementary files

- Supplementary file 1. Table of yeast strains and their genotypes used in this study.
- Supplementary file 2. Table of plasmids and DNA sequences used in this study.
- Transparent reporting form

### Data availability

Data Availability: Sequencing data has been deposited at NCBI under SRA accession #PRJNA573112. Source data files have been provided for Figures 3, 4, and 5.

The following dataset was generated:

| Author(s) | Year | Dataset title | Dataset URL | Database and Identifier |
|---|---|---|---|---|
| Schlichter A, Kasten MM, Parnell TJ, Cairns BC | 2020 | Specialization of the Chromatin Remodeler RSC to Mobilize Partially-Unwrapped Nucleosomes | https://www.ncbi.nlm.nih.gov/sra/PRJNA573112 | NCBI Sequence Read Archive, PRJNA573112 |

The following previously published datasets were used:

| Author(s) | Year | Dataset title | Dataset URL | Database and Identifier |
|---|---|---|---|---|
| Kubik S, O'Duibnir E, Jonge WJ, Mattarocci S, Alber B, Falcone JL, Bruzzone J, Holstege FCP, Shore D | 2015 | NORAP02U | https://www.ncbi.nlm.nih.gov/geo/query/acc.cgi?acc=GSM1891202 | NCBI Gene Expression Omnibus, GSM1891202 |
| Kubik S, O'Duibnir E, Jonge WJ, Mattarocci S, Alber B, Falcone JL, Bruzzone J, Holstege FCP, Shore D | 2015 | NORAP4U | https://www.ncbi.nlm.nih.gov/geo/query/acc.cgi?acc=GSM1891207 | NCBI Gene Expression Omnibus, GSM1891207 |
| Knight B, Kubik S, Ghosh B, Bruzzone MJ, Geertz M, Martin V, Denervaud N, Jacquet P, Ozkan B, Roughemont J, Maerkl SJ, Naef F, Shore D | 2014 | Hmo1 | https://www.ncbi.nlm.nih.gov/geo/query/acc.cgi?acc=GSM1509041 | NCBI Gene Expression Omnibus, GSM1509041 |

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
