## [Decision Letter]

**Acceptance summary:**

It is important to understand the differences between closely related chromatin remodeling complexes, and your studies of the two yeast RSCs provides interesting insights into this question. The combination of in vivo and in vitro studies used here make for a strong story.

**Decision letter after peer review:**

[Editors’ note: the authors submitted for reconsideration following the decision after peer review. What follows is the decision letter after the first round of review.]

Thank you for submitting your work entitled "Specialization of the chromatin remodeler RSC to mobilize partially-unwrapped nucleosomes" for consideration by *eLife*. Your article has been reviewed by three peer reviewers, one of whom is a member of our Board of Reviewing Editors, and the evaluation has been overseen by a Reviewing Editor (me) and a Senior Editor (Kevin Struhl). The following individual involved in review of your submission has agreed to reveal their identity: Blaine Bartholomew (Reviewer #1).

Based on the individual reviews (below) and then further consultations, we regret to inform you that your paper was not accepted for publication in *eLife*. Everybody appreciated the work that went into isolating and analyzing the different RSC variants. There were some concerns about interpretation and quantitation of the in vitro remodeling experiments, although those concerns might be addressed with a little additional work. However, the main criticism was voiced by reviewer 2, who felt that there was a disconnect between the in vitro and in vivo experiments, and therefore that the conclusion that Rsc1 was specialized for partially unwrapped nucleosomes was not strongly justified. After consultation between the reviewers and editors, we felt that these issues would be difficult to address within the window that *eLife* sets for revisions. We hope that these reviews will help you with a future version of the paper.

Reviewer #1:

The authors have carefully characterized the functional differences between the Rsc1 and Rsc2 forms of the RSC ATP-dependent chromatin remodeler and their interplay with Rsc3/Rsc30, two DNA binding subunits of the RSC complex. They found the key difference to be in the two repeats present at the C-terminus of Rsc1 and Rsc2 with the repeat closest to the end helping to regulate the association of Rsc3/30 to the complex. The remodeling differences were only observed between the RSC1 and RSC2 complexes with 5S DNA nucleosomes and not with 601, which suggested the difference might reside in the ability to handle nucleosomes that could be partially unwrapped from the beginning. The combination of biochemistry and genetic experiments are powerful and complementary in this study. However, some of the biochemical experiments were not as well designed as needed to effectively compare the enzymatic differences between the two enzymes or to ascertain as to the experimental reproducibility. The quantitation of nucleosome remodeling assays like that shown in Figure 2C with their corresponding replicates is not displayed and would be clearer if plotted in a line-graph with standard deviation shown for each time point rather than displayed for only of the replicates as a histogram. As it is currently, it is not possible to know what the deviation was like between the replicates. It also seems that they if properly quantitated the authors would also be able to determine the rates of nucleosome movement. The ATPase assays also don't seem to be as effective for comparing difference as they only measure Vmax and measure the extent of ATPase hydrolysis at only one time point with a high concentration of ATP. In their approach it would be easy to miss if there were indeed differences in their ability to hydrolyze ATP. The binding assays are also deficient in that there are only two titration points, there is no quantitation of this data and thus it is also not possible to ascertain as to the standard deviation between the technical replicates similar to the remodeling assays. I was also concerned since remodeling with the 5S DNA nucleosome seems to be generally less efficient than expected and raises questions as to why this might be. Could the linker DNA be too short in the 5S nucleosomes for the Rsc3/30 to bind and be the reason why the addition of Rsc3/30 is inhibitory when based on the genetic data is expected to stimulate the activity of the complex?

They use a powerful approach with a series of histone point mutations to identify histone region(s) that might contribute to the enzymatic differences between the RSC1 and RSC complexes and observed that mutations in the H3 α N helix are synthetically lethal when only RSC2 was present. These are very complementary observations and provide an orthogonal approach to indicate it is the high propensity of unwrapping DNA from the end of the nucleosome that is crucial for these differences. I did however find it odd that in the context of the 5S nucleosomes that H3 R40A mutation appears to weaken these interactions (Figure 3—figure supplement 3); whereas in the context of the 601 nucleosome the same mutation appears to stabilize these interactions over wild type (Figure 4—figure supplement 2 panel B). Does this mean that the destabilizing effect observed with H3 R40A is nucleosome specific and may vary depending on the DNA sequence and would this complicate the authors interpretation? The experiments in Figure 3E indicate that both RSC1 and RSC2 are inhibited by DNA unwrapping from the nucleosome surface, but to different extents. Is it a problem that DNA unwrapping is inhibitory for both forms of RSC and wouldn't it change some of the later interpretations?

The experimental setup in which Hmo1 is added is not adequate to effectively compare how much Hmo1 stimulates the RSC1 versus the RSC2 complex because only 1 time point is used for both enzymes in the remodeling assays. In essence remodeling with RSC1 is much further along because of the time window used, while RSC2 is not and as such you are not able to determine the difference in the stimulation of remodeling when Hmo1 is added. Also, I wasn't clear how an HMG-like protein would help stabilize the wrapping of DNA onto the nucleosome?

The MNase ChIP-seq was a good addition as it helped re-examined the potential differences in the genome-wide Rsc1 and Rsc2 localization and the difference is evident at the tRNA genes and ribosomal protein genes. The RNA-seq also aids in this regard and combined with the other experiments provides a compelling story.

Reviewer #2:

In this manuscript, Cairns and colleagues investigate the potential for in vitro and in vivo distinctions between two forms of the RSC remodeling enzyme that contain either the Rsc1 or Rsc2 subunit. The first half of the manuscript describes extensive biochemical studies with mononucleosomes, comparing the activities of RSC1 and RSC2 complexes. The authors report that the RSC1 complex remodels a 5S nucleosome substrate more efficiently than the RSC2 complex, and that the RSC2 complex has greater problems remodeling a 5S nucleosome harboring a H3-R40A substitution derivative. H4-R40 is located in the H3 N-helix, and the authors find nice synthetic lethality phenotypes between many substitution derivatives in the H3-N helix and a rsc1 deletion strain. Together the data suggest that RSC2 may have a general problem (compared to RSC1) remodeling 5S nucleosomes with alterations in the H3-N helix. Previous work from the Owen-Hughes group has reported that these H3-N helix substitutions lead to increased breathing (spontaneous unwrapping) of entry/exit DNA and increased nucleosome thermal mobility. The authors suggest that RSC1 may be specialized to remodel partially unwrapped nucleosomes. Consistent with this view, they how that RSC1 and RSC2 are proficient at remodeling a more stably wrapped, 601 nucleosome. These in vitro studies lead the authors to predict that RSC1 may play a larger function at genes where RSC is known to engage partially unwrapped nucleosomes, specifically the ribosomal protein (RP) genes. To this end, the carry out extensive ChIP-seq analyzes of Rsc1 and Rsc2, as well as RNA-seq analyzes in rsc1 and rsc2 deletion strains. Contrary to the model, these data reinforce the view that Rsc1 and Rsc2 are functionally redundant at the majority of target genes, including RP genes. The only large different is that Rsc1 appears to play a dominant role at rDNAs. Notably, rDNAs are not known to contain alternative nucleosome structures that would link these functional data with the biochemistry. In general, this work illustrates differences between RSC1 and RSC2 complexes for remodeling a 5S mononucleosome substrate (but not a 601), distinct genetic interactions with histone substitutions, but no clear functional correlation of these results with gene targets or unwrapped nucleosomes in vivo.

1) One major concern with the biochemistry data is the remodeling results with the 5S substrates. This substrate appears to be a center-positioned nucleosome (10N16), but after remodeling, the "slid" product migrates much slower on the native gels. This is counter to every other published sliding assay for RSC or SWI/SNF-like enzymes. These enzymes always move nucleosomes towards or off DNA ends. These always lead to faster migrating species on native gels. For examples, see work from Owen-Hughes with the MMTV A nucleosome (Ferreira et al., 2007) or the 601 nucleosome shown here. Given the mobility of this remodeled species, it is more likely that this product represents a histone dimer loss event, leading to a hexosome or tetrasome. Alternatively, perhaps it is some type of product where the nucleosome is slid off the end and free DNA has re-captured an exposed histone surface (Kassabov et al., 2003). It is unclear why the authors see such a product specifically with a 5S nucleosome, as faster migrating species have been observed with this element previously (Jaskeliof et al., 2000), and early DNase I footprinting assays using the same template shown here gives rise to a "naked DNA" pattern after RSC remodeling, consistent with the nucleosome moved off the DNA ends (Cairns et al., 1996). The identity of this novel species is important for interpreting the data for how histone substitutions or Rsc3/Rsc30 impacts this novel reaction. It also leads to worries that this particular substrate is not representative of typical remodeling events.

2) The authors interpret the impact of H3-N helix substitutions as due to their impact on nucleosome unwrapping. They also argue that 5S is distinguished from 601, based on unwrapping dynamics. It is important to realize that the unwrapping phenotypes were determined by the Owen-Hughes lab using an MMTV A nucleosome, and in some cases also a 601. Note that a H3-K56Q substitution has a large impact on unwrapping a 601 (Neumann et al., 2009). Furthermore, on an MMTV A nucleosome (and/or a 601 nuc), these substitutions stimulate the remodeling activity of the RSC2 complex. This is certainly true for H3-R40A and for H3-K56Q. Indeed, the Owen-Hughes group has argued that RSC2 activity is stimulated by histone substitution derivatives that enhance unwrapping and intrinsic thermal mobility. This is completely opposite of the conclusions and models described here.

3) Figure 4C. This is not an optimal experiment to measure interactions between Hmo1 and RSC in vivo. Sonication of chromatin may yield an average size of ~250-500bp, but the distributions are normally large. This experiment should at least be performed with Mnase trimmed cores. Most likely this will not work, as previous work has shown that Hmo1 needs linker DNA to bind to mononucleosomes (Hepp et al., 2014).

4) Figure 3—figure supplement 3. These mapping data for the 5S mononucleosomes seem odd. Historically, the 5S rRNA gene is not considered a perfect nucleosome positioning sequence. Previous mapping data has indicated that the major translational position only reflects 50% of the population, with the remaining nucleosomes offset by 10 bp intervals. It is surprising then that the mapping work yields a unique right boundary at position 158. Could the "unwrapped" left boundaries really be alternate translational frames, and the right boundaries that might be positioned at the full-length ends (position 174) were not included? It also seems odd that only the left end is "unwrapped" by the H3-R40A substitution.

5) The general interest of this work relies on what is stated in the title that RSC1 is specialized to remodel partially unwrapped nucleosomes and such nucleosome remodeling specialization is important for a particular gene target, such as RP genes. Unfortunately, the in vivo data do not support this model. All of the results show that Rsc1 and Rsc2 share a nearly identical binding profile (with the exception of tDNA), and single deletion strains have nearly no impact on RNA levels, especially RP genes. Furthermore, it is very odd that the only mutant that impacts RP gene expression is the rsc2 rsc30 double mutant. In contrast, the rsc1 rsc30 has little impact. In a simple interpretation, this suggests that Rsc2 is sufficient to support RP gene expression, even without Rsc30. This is not consistent with the main conclusions of the biochemistry or the title.

Reviewer #3:

In this paper, Schlichter et al., compare the Rsc1 and Rsc2 versions of RSC remodeling complex. They find that Rsc2 complexes have reduced affinity for Rsc3/30, and map this difference to the CT2 domain. While the four different complexes (Rsc1 or Rsc2, +/- Rsc3/30) have similar affinity for nucleosomes and ATPase activity, the Rsc1 complexes have stronger nucleosome remodeling activity. They screened histone mutant libraries and found specific synthetic lethality with rsc1 δ in H3 alphaN domain or H3 K56Q (but not A) mutants. These differences correlate with partial nucleosome unwrapping (citing an earlier FRET study). Interestingly, the H3 alphaN synthetic lethality is suppressed if Rsc1 CT2 domain swapped into Rsc2. Using in vitro assays and ChIP experiments the authors build a case that Rsc1 functions preferentially at promoters with wide NDRs (particularly ribosomal protein genes and tDNAs), where partially unwrapped nucleosomes may be more prevalent. Overall, the combination of in vitro biochemistry and correlational in vivo work seems persuasive. The paper is well written and the data relatively clear. Understanding the functional differences between the closely related RSC complexes (or other remodeler families) is of significant interest to the field.

Some specific questions and comments:

1) Does swapping the CT2 domain of Rsc1 into Rsc2 also suppress the synthetic lethality with K56Q?

2) Subsection “RSC2 complexes are deficient in remodeling partially-unwrapped nucleosomes”: The authors conclude that Rsc1 complex is more active for remodeling unwrapped nucleosomes because it has more activity on the R40A nucleosomes than Rsc2. But that is also true for WT 5S nucleosomes. Given that R40A is unlikely to completely disrupt wrapping, isn't it possible that remodeling only occurs when the nucleosomes are wrapped, and R40A simply reduces the time spent in the wrapped conformation? This idea seems supported by the fact that R40A has less effect in the 601 DNA sequence. Following this line of reasoning, could Rsc1 do a better job in inducing unwrapped nucleosomes to adopt the wrapped "remodelable" configuration? The discussion seems to be moving in this direction, but it this is the authors' model it could be made clearer and perhaps foreshadowed more strongly in the results.

[Editors’ note: further revisions were suggested prior to acceptance, as described below.]

Thank you for submitting your article "Specialization of the Chromatin Remodeler RSC to Mobilize Partially-Unwrapped Nucleosomes" for consideration by *eLife*. Your article has been reviewed by three peer reviewers, one of whom is a member of our Board of Reviewing Editors, and the evaluation has been overseen by Kevin Struhl as the Senior Editor. The following individual involved in review of your submission has agreed to reveal their identity: Blaine Bartholomew (Reviewer #3).

The reviewers appreciated the new experiments and additional statistical analysis done in response to their comments. I'm happy to say they all agree that the revised paper will be appropriate for *eLife* with just a few additional changes to the text. This letter is to help you prepare a revised submission.

The reviewers state that this paper has several key results that are solid and significant:

1) The RSC1 enzyme has evolved to use the Rsc3/30 module, and this module facilitates expression of RP genes and helps to counteract histone substitution alleles. Surprisingly, this module has an inhibitory impact in vitro on both enzymes.

2) Hmo1 functions together with RSC1 and RSC2 (though more so with RSC2) to facilitate nucleosome sliding in vitro and gene expression.

3) Rsc1 and Rsc2 show distinct genetic interactions with histone residues.

4) The functional interactions between Rsc1, Rsc2, Hmo1, and Rsc3/30 are enlightening.

5) RSC1 remodels a 5S nucleosome much better than RSC2, though they show more equivalent activity on a 601 nucleosome.

However, the one concern that remains is whether the paper over-simplifies things by making the strong, generalized conclusion (stated in the title and at various points in the paper) that the differential activity of RSC1 and RSC2 is linked to partial unwrapping of the nucleosome. Although the genetic evidence is consistent with your model, reviewer 2 (supported by reviewer 3) cites published data saying the difference in 5S versus 601 unwrapping is not as clear as suggested in your text:

"Introduction. The authors cite 4 papers in support of the statement that "entry/exit DNA displays higher level of detachment from the octamer" (for 5S in comparison to 601). However, none of these papers compare 5S to 601. Polach and Widom, 1995 used 5S, and Li and Widom, 2004 used 601. In the cited review (Zhou et al., 2019), they discuss in detail the comprehensive study by the Poirier group (North et al., 2012; not cited) where they do directly compare 5S and 601 nucleosomes in FRET-based unwrapping assays. Contrary to expectations, a 5S nucleosome unwraps less than 601 at the edge. This is shown to be due to the first 7bp of the 5S sequence at the entry/exit site. Thus, even though 601 has a higher overall affinity, the unwrapping dynamics are higher than 5S. These data are also consistent with published work showing that the 5S sequence has a higher affinity for H2A/H2B dimers (cited in Zhou et al., 2019). Notably, the higher affinity of 601 for octamers is due to AT repeats surrounding the dyad (and contacting the H3/H4 tetramer), not really the edges. As discussed by the authors, differences in how they interpret their data and other work (e.g., older Owen-Hughes work on α N-helix substitutions) may be due to the differences in histone octamer source – yeast vs *Xenopus*. Perhaps yeast dimers have a higher affinity for 601, rather than 5S. However, without extensive discussion, the authors' current conclusions seem to contradict several previously published papers. In general, the remodeling assays don't support a simple model where RSC1 remodels unwrapped nucleosomes better than RSC2. Note that I am not questioning the data at all – they are solid and very well done. I worry that the conclusions may just not be so simple."

In your revised manuscript, please address this point. I took a look at the paper cited by the reviewer (North et al., 10215) and it does seem to claim that 601 unwraps its ends more that 5S. If the reviewer's summary of the literature is accurate, please modify your statements (and model Figure 6) concerning 5S versus 601 accordingly, and factor that into your discussion and interpretations. If you feel those results can be reconciled, or are not applicable, to your model, please explain why in your Discussion section.

---

## [Author Response]

[Editors’ note: the authors resubmitted a revised version of the paper for consideration. What follows is the authors’ response to the first round of review.]

We carefully considered the reviewer comments and your editorial comments – and over the last several months we successfully addressed the vast majority, including those that were indicated as the most important. These efforts have substantially improved our study.

Reviewer #1:The authors have carefully characterized the functional differences between the Rsc1 and Rsc2 forms of the RSC ATP-dependent chromatin remodeler and their interplay with Rsc3/Rsc30, two DNA binding subunits of the RSC complex. They found the key difference to be in the two repeats present at the C-terminus of Rsc1 and Rsc2 with the repeat closest to the end helping to regulate the association of Rsc3/30 to the complex. The remodeling differences were only observed between the RSC1 and RSC2 complexes with 5S DNA nucleosomes and not with 601, which suggested the difference might reside in the ability to handle nucleosomes that could be partially unwrapped from the beginning. The combination of biochemistry and genetic experiments are powerful and complementary in this study. However, some of the biochemical experiments were not as well designed as needed to effectively compare the enzymatic differences between the two enzymes or to ascertain as to the experimental reproducibility. The quantitation of nucleosome remodeling assays like that shown in Figure 2C with their corresponding replicates is not displayed and would be clearer if plotted in a line-graph with standard deviation shown for each time point rather than displayed for only of the replicates as a histogram. As it is currently, it is not possible to know what the deviation was like between the replicates. It also seems that they if properly quantitated the authors would also be able to determine the rates of nucleosome movement.

We agree that additional replicates and display modes would improve the manuscript. We have quantified replicates of the sliding assays as the reviewer recommends and these are now included as additional supplemental figures for each assay. Figure 2—figure supplement 1, Figure 3—figure supplement 3, Figure 4—figure supplement 2, Figure 4—figure supplement 3B, Figure 4—figure supplement 4, and Figure 4—figure supplement 6.

The ATPase assays also don't seem to be as effective for comparing difference as they only measure Vmax and measure the extent of ATPase hydrolysis at only one time point with a high concentration of ATP. In their approach it would be easy to miss if there were indeed differences in their ability to hydrolyze ATP.

We have replaced the ATPase bar graph with an ATPase time course in Figure 2B.

The binding assays are also deficient in that there are only two titration points, there is no quantitation of this data and thus it is also not possible to ascertain as to the standard deviation between the technical replicates similar to the remodeling assays.

We observe strong differences in RSC remodeling between wt 5S and H3R40A 5S yNucs using nuc:RSC ratios of 1:1.5. Our nucleosome binding assays were conducted at nuc:RSC ratios of 1:1.5, as well as 1:3. Since the nuc binding assay utilizes the conditions used in the remodeling assay, this indicates that RSC1 and RSC2 bind to wt and R40A 5S yeast nucleosomes similarly under these conditions. Additional titration points would require increasing the amount of RSC complex in the assay relative to the amount of nucleosome. Currently this would be challenging because the remodeler concentration required for these higher titration points would be difficult to obtain while remaining within assay parameters for a crisp RSC:Nuc gelshift band. In addition, the higher remodeler concentration would be needed for all four RSC complexes, which would require four new preps which are not feasible to produce at this time. However, we have included the technical replicates for the current experiments as support in the Supplementary Figure.

I was also concerned since remodeling with the 5S DNA nucleosome seems to be generally less efficient than expected and raises questions as to why this might be. Could the linker DNA be too short in the 5S nucleosomes for the Rsc3/30 to bind and be the reason why the addition of Rsc3/30 is inhibitory when based on the genetic data is expected to stimulate the activity of the complex?

We note that we do conduct our work under catalytic conditions, with excess substrate. Regarding length, we have lengthened 5S NPS to 198bp with similar, though more complicated, remodeling results because this NPS resulted in 2 different nucleosome ‘slid’ positions. However, as there was no striking difference with the Rsc3/30 in vitro effect (same result as with shorter DNA), we have not presented experiments using this longer NPS in the paper. Furthermore, it is clear RSC sliding is much more efficient on the 5S nucleosome in the presence of Hmo1, which we discuss in depth in the paper.

They use a powerful approach with a series of histone point mutations to identify histone region(s) that might contribute to the enzymatic differences between the RSC1 and RSC complexes and observed that mutations in the H3 α N helix are synthetically lethal when only RSC2 was present. These are very complementary observations and provide an orthogonal approach to indicate it is the high propensity of unwrapping DNA from the end of the nucleosome that is crucial for these differences. I did however find it odd that in the context of the 5S nucleosomes that H3 R40A mutation appears to weaken these interactions (Figure 3—figure supplement 3); whereas in the context of the 601 nucleosome the same mutation appears to stabilize these interactions over wild type (Figure 4 —figure supplement 2 panel B). Does this mean that the destabilizing effect observed with H3 R40A is nucleosome specific and may vary depending on the DNA sequence and would this complicate the authors interpretation?

Indeed, the H3 R40A mutation causes further unwrapping of the 5S nucleosome, whereas it has little effect on a 601 nucleosome. We agree that the destabilizing effect is likely nucleosome specific depending on DNA sequence, and that the effect is greater with a DNA sequence that has a lesser affinity for the octamer. We would expect the greatest effect on further destabilization to be at regions with sequences that are inherently partially unwrapped (similar to 5S) such as those found at RPGs and other wide NDRs (“fragile/MNase sensitive nucleosomes”).

We note that reviewer 3’s comments (see below) for grouping the wrapped and unwrapped products together and listing percentages for the individual species was indeed helpful, will assist in the interpretation here. In keeping, we have made these changes to Figure 3—figure supplement 5 and Figure 4—figure supplement 5.

The experiments in Figure 3E indicate that both RSC1 and RSC2 are inhibited by DNA unwrapping from the nucleosome surface, but to different extents. Is it a problem that DNA unwrapping is inhibitory for both forms of RSC and wouldn't it change some of the later interpretations?

The reviewer is correct, and we needed to make this point more clearly in our revision, as two reviewers had questions here. To clarify, our data and model are indeed that wrapped nucleosomes are better substrates for both RSC1 and RSC2 complexes, but RSC1 can remodel the unwrapped version better than RSC2 – especially when unaided by other proteins. We have now stated this clearly in two locations in the revision – both in the Results section and in the Discussion section. Furthermore, our data supports RSC complex as two redundant functional entities that manage partially-wrapped nucleosomes: ‘Rsc2 with Hmo1’ and ‘Rsc1 with Rsc3-30’ – and one must significantly impair both entities for a strong phenotype (see below).

The experimental setup in which Hmo1 is added is not adequate to effectively compare how much Hmo1 stimulates the RSC1 versus the RSC2 complex because only 1 time point is used for both enzymes in the remodeling assays. In essence remodeling with RSC1 is much further along because of the time window used, while RSC2 is not and as such you are not able to determine the difference in the stimulation of remodeling when Hmo1 is added.

We agree, and for this reason different time point and remodeler concentrations are shown in supplemental Figure 4—figure supplement 3, so that the “fast” remodelers are tested when they aren’t “further along.” These results reinforce our conclusions.

Also, I wasn't clear how an HMG-like protein would help stabilize the wrapping of DNA onto the nucleosome?

Hmo1 has been shown to stabilize chromatin and perform the functions of a linker histone. Unlike other HMGB proteins which have an acidic CTD that promotes bending, Hmo1 is relatively unique in this class as it contains a basic lysine rich C-terminal extension. Hmo1 is predicted to bind near the nucleosome dyad and use its basic extension to bind linker DNA and prevent bending and improve wrapping. Additionally, nucleosomes in an *hmo1* null display increased MNase sensitivity. (Panday and Grove, 2016; Panday and Grove, 2017). We have incorporated additional text in the manuscript to convey this to the reader.

The MNase ChIP-seq was a good addition as it helped re-examined the potential differences in the genome-wide Rsc1 and Rsc2 localization and the difference is evident at the tRNA genes and ribosomal protein genes. The RNA-seq also aids in this regard and combined with the other experiments provides a compelling story.

We thank the reviewer for appreciating these experiments.

Reviewer #2:[…] 1) One major concern with the biochemistry data is the remodeling results with the 5S substrates. This substrate appears to be a center-positioned nucleosome (10N16), but after remodeling, the "slid" product migrates much slower on the native gels. This is counter to every other published sliding assay for RSC or SWI/SNF-like enzymes. These enzymes always move nucleosomes towards or off DNA ends. These always lead to faster migrating species on native gels. For examples, see work from Owen-Hughes with the MMTV A nucleosome (Ferreira et al., 2007) or the 601 nucleosome shown here. Given the mobility of this remodeled species, it is more likely that this product represents a histone dimer loss event, leading to a hexosome or tetrasome. Alternatively, perhaps it is some type of product where the nucleosome is slid off the end and free DNA has re-captured an exposed histone surface (Kassabov et al., 2003). It is unclear why the authors see such a product specifically with a 5S nucleosome, as faster migrating species have been observed with this element previously (Jaskeliof et al., 2000), and early DNase I footprinting assays using the same template shown here gives rise to a "naked DNA" pattern after RSC remodeling, consistent with the nucleosome moved off the DNA ends (Cairns et al., 1996). The identity of this novel species is important for interpreting the data for how histone substitutions or Rsc3/Rsc30 impacts this novel reaction. It also leads to worries that this particular substrate is not representative of typical remodeling events.

This is a highly relevant point, and we thus thought it important to quantitatively assess whether the slower migrating species in the sliding assay could be due to dimer loss during RSC remodeling. We addressed this issue directly by examining 5S sliding on fluorescently-labeled yeast nucleosomes. Here, we assembled 174bp 5S yeast nucleosomes labeled with Oregon Green at Q114C of H2A, and used them in RSC sliding reactions. RSC1 complex remodeled these labeled nucleosomes resulting in a slower migrating band, as we have seen with remodeled unlabeled 5S nucleosomes. If H2A/H2B dimer was lost from the slid product during remodeling, then we would expect a reduction in the dimer to DNA ratio when normalized to the ratio observed in the starting nucleosomes. However, the dimer to DNA ratio does not decrease in the slid product, and instead is very similar to the starting nucleosomes. This data has been added to the paper as Figure 2—figure supplement 3.

The reviewer states that all nucleosomes positioned near or off of DNA ends will always lead to faster, not slower, migrating species on native gels. A complex’s migration through a native gel is affected by its overall shape as well as charge. It is known that different DNA sequences have different degrees of inherent stiffness and flexibility. It seems reasonable that some DNA sequences protruding from a nucleosome could alter its migration on a native gel. This would be a property of the specific NPS and nucleosomes. For example, Chakravarthy et al., (2012) (Bowman lab) reported remodeled nucleosomes that ran aberrantly fast on native gels which showed a smaller effective size for the particle and greater compaction.

After examining, we did not consider the mononucleosome sliding results of the 5S template used in the Jaskelioff, 2000 paper as comparable to our 5S mononucleosome sliding. The Jaskelioff template is 416 bp with 2 copies of the 5S NPS (compatible with di-nucleosome formation and examination) while ours is a 174 bp fragment with a single 5S NPS, for mononucleosome formation and examination. Thus, it seems entirely possible that these two templates would migrate differently on native gels before or after remodeling.

Although we believe our Oregon Green labeling has resolved the H2A/B dimer issue for our experiments, we will mention that some hexasomes migrate slower than nucleosomes (Chen et al., 2017), while others run faster (Levendosky et al., 2016, Brehove et al., 2019, Mazurkiewicz et al., 2006). Indeed, the Mazurkiewicz, 2006 paper shows Sea Urchin 5S hexasomes and nucleosomes on 146 bp and 207 bp NPS fragments and their experiments show hexasomes running faster than the corresponding nucleosome.

Regarding the early 5S DNase I footprinting after RSC remodeling (Cairns, 1996), where the product resembles naked DNA. I agree with the reviewer that this observation is consistent with dimer loss – reinforcing the need for conducting the Oregon Green experiment. As the dimers are shown to remain, I believe that RSC has moved these nucleosomes out of their initial/preferred rotational phase to a superposition of all rotational phases, and thus the DNaseI patterns will resemble naked DNA – with the fraction of end position nucleosomes lacking (as the reviewer notes) octamer protection.

2) The authors interpret the impact of H3-N helix substitutions as due to their impact on nucleosome unwrapping. They also argue that 5S is distinguished from 601, based on unwrapping dynamics. It is important to realize that the unwrapping phenotypes were determined by the Owen-Hughes lab using an MMTV A nucleosome, and in some cases also a 601. Note that a H3-K56Q substitution has a large impact on unwrapping a 601 (Neumann et al., 2009). Furthermore, on an MMTV A nucleosome (and/or a 601 nuc), these substitutions stimulate the remodeling activity of the RSC2 complex. This is certainly true for H3-R40A and for H3-K56Q. Indeed, the Owen-Hughes group has argued that RSC2 activity is stimulated by histone substitution derivatives that enhance unwrapping and intrinsic thermal mobility. This is completely opposite of the conclusions and models described here.

There are indeed apparent differences with prior results, but we believe they can be reconciled. First, the type of NPS sequence and DNA length are important factors for nucleosome wrapping and H3-α N helix mutations may behave differently in the context of the Sea Urchin 5S nucleosomes examined here than in the context of other NPS previously studied. More to the point, we consider it reasonable and likely that histone origin (e.g., yeast, *Drosophila*, *Xenopus*, or chicken) influences nucleosome wrapping, and underlies differences in observations. Our experiments utilize yeast remodeler with yeast octamers (nucleosomes), whereas the papers cited (and the majority in the field) use a variety of octamer sources for the in vitro work, but rarely have people used yeast remodelers with yeast nucleosomes. Our lab has observed differences in remodeler activities based on octamer source, and therefore we elected to use only yeast octamers to test yeast RSC derivatives, as we believe this provides a higher degree of confidence in the results.

We mapped the positioning of the wild-type and mutant recombinant nucleosomes used in this study, which supports the hypothesis that canonical Sea Urchin 5S sequence with yeast nucleosomes are less well wrapped than original Widom 601 yeast nucleosomes, and that RSC1 and RSC2 behave differently on these two templates. The H3 R40A mutation was not sufficient to unwrap the yeast Widom 601 nucleosome nor did it affect RSC sliding on yeast 601 nucs in our experiments.

Importantly, while the Owen-Hughes group does indeed show an increase in the initial rate of repositioning (1.2-2x -measured as the sum of all remodeled products) by RSC for the same H3 mutations where we see lethality with *rsc1∆* (e.g., H3R40A; subsection “Well-wrapped nucleosomes are remodeled comparably by both RSC1 and RSC2 complexes”). Importantly, they also show that when these mutant nucleosomes are remodeled by RSC2 and run on a native polyacrylamide gel, they show significantly altered products compared to WT – so the distribution of products is highly affected. So, while there may be aspects of remodeling that are enhanced, proper remodeling to completion appears impaired (see subsection “Well-wrapped nucleosomes are remodeled comparably by both RSC1 and RSC2 complexes”). This improper remodeling by RSC2 of R40A is therefore supported by our in vivo genetic studies.

3) Figure 4C. This is not an optimal experiment to measure interactions between Hmo1 and RSC in vivo. Sonication of chromatin may yield an average size of ~250-500bp, but the distributions are normally large. This experiment should at least be performed with Mnase trimmed cores. Most likely this will not work, as previous work has shown that Hmo1 needs linker DNA to bind to mononucleosomes (Hepp et al., 2014).

This is a good point, and our DNA fragment size following sonication using the Bioruptor Pico is predominantly 150-300 bp. In addition, we have now repeated the experiment from crosslinked MNase-treated chromatin extracts as suggested by the reviewer and still see co-immunoprecipitation. This is included as Figure 4—figure supplement 1.

4) Figure 3—figure supplement 3. These mapping data for the 5S mononucleosomes seem odd. Historically, the 5S rRNA gene is not considered a perfect nucleosome positioning sequence. Previous mapping data has indicated that the major translational position only reflects 50% of the population, with the remaining nucleosomes offset by 10 bp intervals. It is surprising then that the mapping work yields a unique right boundary at position 158. Could the "unwrapped" left boundaries really be alternate translational frames, and the right boundaries that might be positioned at the full-length ends (position 174) were not included? It also seems odd that only the left end is "unwrapped" by the H3-R40A substitution.

We used paired-end sequencing after ExoIII and S1 digestion to map the recombinant yeast nucleosomes and agree the 5S is not a perfect NPS. Paired-end sequencing gives fragment length, start and end points for each individual fragment, and number of reads indicating strength of fragment representation, which we provide. We have improved the display of our mapping data as suggested by reviewer 3. Our wild type 5S nucleosome mapping is in agreement with the asymmetric unwrapping previously shown for the 5S NPS by Winogradoff and Aksimentiev, 2019).

5) The general interest of this work relies on what is stated in the title that RSC1 is specialized to remodel partially unwrapped nucleosomes and such nucleosome remodeling specialization is important for a particular gene target, such as RP genes. Unfortunately, the in vivo data do not support this model. All of the results show that Rsc1 and Rsc2 share a nearly identical binding profile (with the exception of tDNA), and single deletion strains have nearly no impact on RNA levels, especially RP genes. Furthermore, it is very odd that the only mutant that impacts RP gene expression is the rsc2 rsc30 double mutant. In contrast, the rsc1 rsc30 has little impact. In a simple interpretation, this suggests that Rsc2 is sufficient to support RP gene expression, even without Rsc30. This is not consistent with the main conclusions of the biochemistry or the title.

I believe these comments are a result of our inadequate description of our interpretations and model. First, we agree that (beyond tDNAs) the binding profiles of Rsc1 and Rsc2 are nearly identical. That fact was stated in the manuscript and consistent with our model, but we did not do an adequate job stating the model. To rectify, we now have sections in the expanded Discussion section on the issue of how RSC1 complexes better manage partially-unwrapped nucleosomes after (independent of) the initial nucleosome binding step. Therefore, if the initial binding is not affected, then the genomics should show them both occupying the same Pol II genes.

Second, we clarify in key locations how the in vitro and in vivo data are largely consistent, and that although Rsc1/2 are at the same locations, they can have partially overlapping/redundant, and partially unique functions – summarized in Figure 5—figure supplement 2 and in the Model figure (Figure 6). Put succinctly, our data supports RSC as two partially redundant functional entities: ‘Rsc2 with Hmo1’ and ‘Rsc1 with Rsc3-30’ – and one must significantly impair both entities for a strong phenotype in vivo (see Figure 5—figure supplement 2C). We note that in vitro tests on particular nucleosome substrates cannot encompass the complexity of genome sequences that RSC manages. However, there is general agreement in the data to support differences in the ability of RSC1 versus RSC2 to manage partially-unwrapped nucleosomes – as the best current model to explain our genetic, biochemical and genomic data.

Regarding RP gene expression in particular, we find that yeast prior to the whole genome duplication do not have Rsc1, and as neither *rsc1* nor *rsc30* nulls are inviable, RSC2 (with Hmo1) must be sufficient to support RP gene expression. The *rsc1*, *rsc2*, and *rsc1 rsc30* mutations have mild effects because of this built in redundancy that Rsc2/Hmo1 can perform the role when *RSC1* is absent. In the *rsc2 rsc30* mutation, RSC2 complex is absent, and the RSC1 complex is now impaired due to loss of *RSC30*, which causes a significant impact on RP expression. As mentioned above, we have included a table depicting the redundancy relationship in Figure 5—figure supplement 2C.

We speculate that Rsc1, as a later evolutionary adaptation, seems to be specialized to help remodel these regions. RSC1 is only 20% of total RSC complexes, yet has a slightly higher occupancy at RPGs, wide NDRs and tDNAs than RSC2. While the differences in occupancy and expression may not appear that substantial as a result of the redundancy, having nucleosomes poised in a specialized state that is “better” handled by one form of RSC provides for rapid and regulated action in response to environmental signals, and likely provides a fitness advantage to yeast in the wild.

Reviewer #3:[…] Some specific questions and comments:1) Does swapping the CT2 domain of Rsc1 into Rsc2 also suppress the synthetic lethality with K56Q?

Yes it does. This is shown in Figure 3—figure supplement 2B.

2) Subsection “RSC2 complexes are deficient in remodeling partially-unwrapped nucleosomes”: The authors conclude that Rsc1 complex is more active for remodeling unwrapped nucleosomes because it has more activity on the R40A nucleosomes than Rsc2. But that is also true for WT 5S nucleosomes. Given that R40A is unlikely to completely disrupt wrapping, isn't it possible that remodeling only occurs when the nucleosomes are wrapped, and R40A simply reduces the time spent in the wrapped conformation? This idea seems supported by the fact that R40A has less effect in the 601 DNA sequence. Following this line of reasoning, could Rsc1 do a better job in inducing unwrapped nucleosomes to adopt the wrapped "remodelable" configuration? The discussion seems to be moving in this direction, but it this is the authors' model it could be made clearer and perhaps foreshadowed more strongly in the results.

Yes, the idea that Rsc1 could do a better job inducing the wrapped conformation, likely via its onboard Rsc3/30 module which could bind the DNA and help “wrap” or stabilize the nucleosome (much like Hmo1) fits nicely within our model. The alternative is that RSC1 complex could be more tolerant of a partially-wrapped nucleosome to initiate DNA translocation. We include both of these possibilities in the new Discussion.

[Editors’ note: what follows is the authors’ response to the second round of review.]

[…] In your revised manuscript, please address this point. I took a look at the paper cited by the reviewer (North et al., 10215) and it does seem to claim that 601 unwraps its ends more that 5S. If the reviewer's summary of the literature is accurate, please modify your statements (and model Figure 6) concerning 5S versus 601 accordingly, and factor that into your discussion and interpretations. If you feel those results can be reconciled, or are not applicable, to your model, please explain why in your Discussion section.

Again, we would like to thank the reviewers and editors for their efforts in reviewing and helpful comments – which improved the manuscript – and for their appreciation of the quality and significance of the work. We were delighted to have the paper provisionally accepted at *eLife*. Below we address the one remaining request from the editors and reviewer.

One reviewer expresses concern that our paper conveys a more simplified conclusion (nucleosome partial unwrapping) than merited for differences between Rsc1 versus Rsc2, given that a prior paper has found that the 601 DNA sequence displays more unwrapping from the nucleosome edge (entry/exit) than does the *5S* sequence (North et al., 2012). The reviewer is right to point out this paper, and we agree that additional discussion would be helpful to reconcile their observations with ours. To fully address, we will discuss a major caveat with the North et al., paper, discuss subsequent published work showing higher unwrapping with *5S* compared to 601, and emphasize that with our nucleosomes (using yeast octamers with either sea urchin *5S* or the Widom 601 sequence), the *5S* displays more unwrapping than does the 601 nucleosome.

First, we utilized the sea urchin *5S* sequence, while the North et al., paper used *Xenopus*, *5S* – so the DNA sequences are slightly different. Second, and much more importantly, North et al. make a significant alteration to both the *5S* and 601 sequences by inserting the LexA binding sequence within the region that unwraps from the octamer. In addition, the authors state that without LexA bound, *5S* has a lower FRET efficiency than 601, possibly due to increased unwrapping by the first few base pairs of the *5S*. Third, there are additional studies, subsequent to North et al. (e.g., Chen et al., 2014 and Mauney et al., 2018) that used small angle X-ray scattering (SAXS) to compare native/unaltered sea urchin *5S* to the Widom 601 sequence, which showed that the *5S* unwraps more rapidly and at lower salt concentrations than does 601. Fourth, our studies utilize yeast octamers to form nucleosomes, which we consider best suited to the study of yeast remodelers, rather than the more common usage of either *Drosophila* or *Xenopus* histones. Here, yeast octamers might behave somewhat differently in their wrapping properties.

Importantly, rather than simply relying on prior studies, and given that our studies use yeast octamers with the *5S* and 601 sequences, we directly examined the extent of unwrapping with the *5S* and 601 nucleosomes used in our studies. To this end, we performed ExoIII-S1 nuclease mapping combined with high throughput sequencing to determine the extent of unwrapping in our starting nucleosomal templates, and found that *5S* nucleosomes are unwrapped to a greater extent than 601 nucleosomes, and that the H3 R40A mutation enhances the unwrapping of the *5S*, but not the 601 template. (Figure 3—figure supplement 5, and Figure 4—figure supplement 5B). Our mapping studies are thus in agreement with the findings from Chen et al., 2014 and Mauney et al., 2018. Taken together with our genetics results, and the positive impact (in vitro and in vivo) of Hmo1, which is predicted to improve wrapping – we conclude that partial nucleosome unwrapping provides the most parsimonious explanation for the differences observed in remodeling capacity between Rsc1 and Rsc2 complexes.

To address this issue in the manuscript, we have added content and these additional references to the Introduction, made a clearer statement about our mapping of the unwrapping of the *5S* and 601 starting templates in the Results section, and added a paragraph to the Discussion section.